# Rat superior colliculus encodes the transition between static and dynamic vision modes

Rita Gil [1,2], Mafalda Valente [1,2] & Noam Shemesh [1] ✉

The visual continuity illusion involves a shift in visual perception from static to dynamic vision modes when the stimuli arrive at high temporal frequency, and is critical for recognizing objects moving in the environment. However, how this illusion is encoded across the visual pathway remains poorly understood, with disparate frequency thresholds at retinal, cortical, and behavioural levels suggesting the involvement of other brain areas. Here, we employ a multi-modal approach encompassing behaviour, whole-brain functional MRI, and electrophysiological measurements, for investigating the encoding of the continuity illusion in rats. Behavioural experiments report a frequency threshold of 18±2 Hz. Functional MRI reveal that superior colliculus signals transition from positive to negative at the behaviourally-driven threshold, unlike thalamic and cortical areas. Electrophysiological recordings indicate that these transitions are underpinned by neural activation/suppression. Lesions in the primary visual cortex reveal this effect to be intrinsic to the superior colliculus (under a cortical gain effect). Our findings highlight the superior colliculus' crucial involvement in encoding temporal frequency shifts, especially the change from static to dynamic vision modes.

The mammalian visual system[1–5] has evolved ingenious ways of recognizing and extracting visual features that enable object perception[5,6] and visual motion detection[7–10], both essential for interacting with the external environment. The encoding of spatial resolution features along the entire visual pathway is well characterized, with most brain structures exhibiting topographical mappings[11–15] that systematically represent the visual space. By contrast, how visual systems resolve luminance changes over time[16,17] has yet to be explained on a systems level, with most studies focusing mainly on the retina[18–20] and/ or the visual cortex (VC)[21–23].

A critical temporal phenomenon for visual perception is the continuity illusion effect: when photons impinge on the retina, the visual pathway can operate in static vision mode—whereby every flash is encoded as a separate event promoting attention and novelty perception—or can shift to the dynamic vision mode, where flashing stimuli are "fused", thereby producing a continuity illusion where light is

perceived as continuous and steady[16,24,25]. This temporal continuity effect is different from, for example, illusory motion[26], which introduces spatial features into the processing of continuity. In the temporal continuity illusion, the flicker fusion frequency (FFF) threshold is typically defined as the frequency at which the transition from static to dynamic vision modes occurs as reported by the specific experimental modality being used, which can lead to ambiguity. In electrophysiology the FFF threshold has been defined as the frequency at which individual flash-evoked potentials can no longer be resolved, while in behavioural essays the threshold is defined based on reports on visual perception. Retinal cyto-organization strongly affects measured FFF thresholds[19,27,28]: For example, diurnal fast-moving animals such as birds possess high visual temporal resolution which enables the detection and processing of fast-moving stimuli, such as prey, obstacles, as well as maintaining formation when flying in flocks[24]. The FFF threshold also plays important roles in prey–predator interactions,

[1]Champalimaud Research, Champalimaud Foundation, Lisbon, Portugal. [2]These authors contributed equally: Rita Gil, Mafalda Valente.
✉e-mail: noam.shemesh@neuro.fchampalimaud.org

for example, in camouflaging moving prey, or in detecting predators (a dynamically changing appearance can elicit a startle/fear response, giving prey an advantage to escape)[29]. Systemic medical conditions such as hepatic encephalopathy or eye disorders such as cataracts or glaucoma can also strongly affect the FFF threshold and thus visual perception[16,24].

Interestingly, FFF thresholds derived from behaviour[21,24,30–34] and electrophysiological recordings[18–23] are disparate. For example, hens do not appear to behaviourally perceive flicker frequencies above 75–87 Hz[24,35], while their electroretinograms (ERGs) remained in phase with the flickering light well beyond 100 Hz[18,19]. Similar trends were observed in mice, where ERGs vs. behavioural reports of FFF thresholds were ~30[18] and ~14 Hz[36], respectively. Strikingly, electrophysiological recordings in the cortical end of the visual pathway disagreed both with behavioural- and ERGs-derived FFF thresholds, further suggesting that behaviourally relevant FFF threshold encoding may occur elsewhere along the pathway.

Here, we combined behavioural measurements, network-level functional MRI (fMRI), and electrophysiological recordings, and cortical lesions, to investigate the network-level neural correlates of the transition from static to dynamic vision modes. We find that mechanisms boosting/suppressing neural activity in the superior colliculus (SC) are strongly associated with behavioural reports of static/dynamic vision modes, suggesting that the SC is a major junction for flicker fusion.

## Results

### Rats behaviourally report shifts from static to dynamic vision modes at 18 ± 2 Hz

We designed a simple psychophysical task to estimate the FFF threshold proxy in rats (Fig. 1 and Supplementary Movie 1). Rats were placed in a box with three ports and trained to report to one side port when the stimulus was continuous, and to the other when the light stimulus was flickering at 2 Hz (counterbalanced across animals; cf. "Methods" section). To ensure exposure to several flashes, even at low frequencies, rats were required to wait for 1 s before a tone signalled that a report was allowed. When animals chose the correct port, a water reward was delivered. Rats could perform this discrimination with an accuracy above 95% (Fig. S1). The percentage of flicker port reports is shown in Fig. 1C (a similar analysis comparing two different adaptation periods is shown in Fig. S2). For frequencies above approximately 20 Hz, rats predominantly chose the port rewarded to continuous stimuli. Notably, as the flicker frequency increased, the percentage of flicker port reports decreased, indicating that the animal tended to choose the continuous light port for higher frequencies. As

the difficulty level for discriminating between individual flashes increased, the animals transitioned towards the dynamic vision mode: for the FFF threshold proxy calculation a sigmoid curve was fitted to the average animal response and the intercept at 0.5 (considered to be "chance level" as animals reported equally to both ports) was taken. The calculated FFF threshold proxy was 18 ± 2 Hz and the confidence intervals were defined via a bootstrap method (cf. "Methods" section).

### Pathway-level fMRI reveals that SC signals, but not cortical or thalamic signals, tightly track behavioural reports

We then turned to investigate activity in the entire visual pathway in a separate group of rats (n = 18) via fMRI experiments conducted at 9.4 T with binocular flashing visual simulation (spatial resolution of ~270 × 270 μm² in-plane, 1.5 mm slice thickness and 1.5 s temporal resolution). Physiological conditions and MRI pulse sequence were optimized (Figs. S3–5). The stimulation paradigm, and LED positioning, relative to the animal's eyes, are shown in Fig. 2A. Functional activation t-maps for representative stimulation frequencies of 1, 15, 25 Hz and continuous light are shown in Fig. 2B. At the lowest stimulation frequency, strong positive blood-oxygenation-level-dependent (BOLD) responses (PBRs) were observed in subcortical structures of the visual pathway (SC and thalamic lateral geniculate nucleus of the thalamus−LGN). Cortical areas exhibit somewhat weaker PBRs. As the stimulation frequency increased, gradual shifts were observed from PBRs to negative BOLD responses (NBRs), first in VC and then in SC. LGN responses remained positive for all frequencies, but t-values decreased with frequency. Temporal profiles in the anatomically defined regions of interest (ROIs) confirmed the trends described above (Fig. 2C) and further revealed sharp positive signals at the beginning and the end of stimulation for the higher frequency stimuli in SC (hereafter referred to as onset and offset signals, respectively) flanking a lower "steady-state" fMRI response.

To investigate the relationship between pathway-wide fMRI responses and behaviour results, we measured a larger stimulation frequency space and correlated the mean fMRI percent signal change across rats during the stimulation "steady-state" (cf. "Methods" section) with the mean probability of reporting to the flicker port across animals (Fig. 2D). Interestingly, VC and LGN evidenced saturated responses: for frequencies above 15 Hz, the negative VC responses remained rather constant, and, in LGN, signals exhibited very small positive percent signal change above 20 Hz.

By contrast, a much broader dynamic range was observed in the SC, where fMRI "steady-state" signals crossed zero between 15 and 20 Hz−close to the behaviourally-measured FFF threshold surrogate. Furthermore, the SC was the only structure that exhibited a

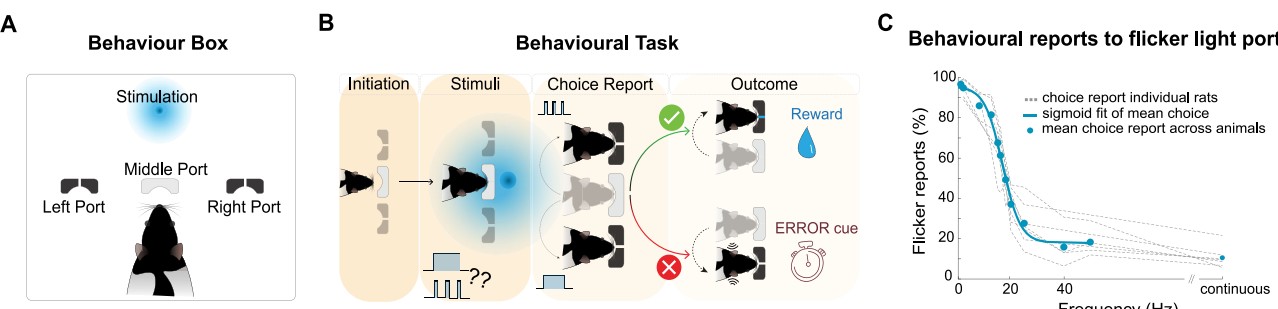

**Fig. 1 | Behavioural results. A** Behaviour box schematic. Water-deprived rats were placed in a dark box with three poking ports: a middle port to initiate trials and two lateral ports for continuous or flicker light reports. **B** Schematic of the task: Animals start each trial by poking in the central port. An overhead LED would turn on displaying either continuous light or flickering light at various frequencies. Rats were rewarded for poking to one side if the stimulus was continuous, and to the other side if the stimulus flickered. Incorrect responses triggered a noise burst and

a time penalty. **C** Percentage of reports to the flicker port. Thin grey dashed lines reflect the performance of each individual animal (n = 7) while blue circles correspond to the averaged individual performances. As the frequency increases the animal reports less often to the flickered port signalling a shift towards the dynamic vision mode. The calculated FFF threshold proxy at "chance level" is 18 ± 2 Hz. Source data are provided as a Source Data file.

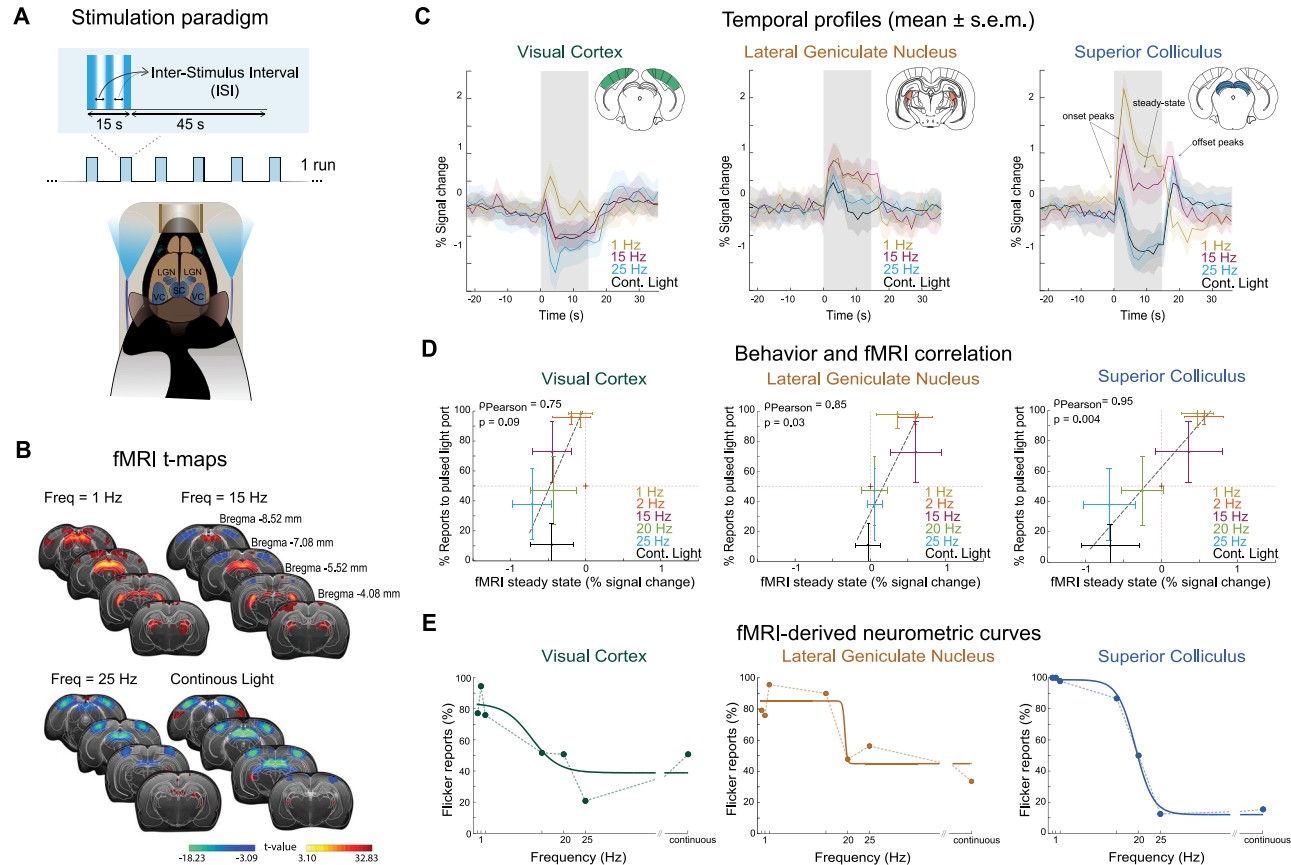

**Fig. 2 | Whole pathway fMRI results (total *n* = 18, for every individual frequency *n* ≥ 5). A** TOP: Stimulation paradigm used in the fMRI. The stimulation paradigm consisted of 15 s stimulation at different frequencies followed by 45 s rest; Bottom: Schematic of the animal's position in MR bed. **B** fMRI t-maps for representative visual stimulation frequencies and atlas overlapped. As the stimulation frequency increases, transitions from PBRs to NBRs are observed first in VC and subsequently in SC. **C** Mean ± s.e.m. of fMRI signal temporal profiles across animals (*n* = 18). Fine structure appears in the fMRI responses. Onset and offset peaks are evident in the SC profiles from 15 Hz onwards, along with a "steady-state" (black arrows). **D** Correlation of behaviour reports with fMRI "steady-state" signals. Coloured circles represent the average response of behavioural and fMRI sessions and error

bars represent the standard deviation across runs/sessions. Only the SC shows a clear transition from PBR to NBR that correlates with the behaviourally measured FFF threshold surrogate (*R*(4) = 0.95, *P* = 0.004, 95% CI = [0.61,1.00], two-tailed Pearson's correlation). Weaker skewed correlations were found for the VC and LGN (*R*(4) = 0.75, *P* = 0.09, 95% CI = [−0.16,0.97] and *R*(4) = 0.85, *P* = 0.03, 95% CI = [0.11, 0.98], respectively, two-tailed Pearson's correlation). **E** fMRI-derived neurometric curves generated from individual trial data reveal that only SC tracks the psychometric curve obtained from the behavioural experiments, reporting a "chance level" threshold of 20.0 ± 2.8 Hz. PBR positive BOLD responses, BOLD blood oxygenated level dependent, NBR negative BOLD responses, VC visual cortex, SC superior colliculus. Source data are provided as a Source Data file.

statistically significant correlation coefficient $\rho_{Pearson}$ (4) = 0.95 (*P* = 0.004, 95% CI = [0.61,1.00], two-tailed Pearson's correlation). Although this correlation is based on mean responses across 6 frequencies, it failed to reach statistical significance in the VC and LGN with $\rho_{Pearson}$ (4) = 0.75 (*P* = 0.09, 95% CI = [−0.16,0.97], two-tailed Pearson's correlation) and $\rho_{Pearson}$ (4) = 0.85 (*P* = 0.03, 95% CI = [0.11, 0.98], two-tailed Pearson's correlation), respectively.

Finally, to quantify the single-trial discriminability of the fMRI responses as a function of flicker frequency, we computed fMRI-derived neurometric curves[37] for each ROI (Fig. 2E). This was done by estimating the proportion of single-trial fMRI percent change responses (averaged across animals) above the mean value for the 20 Hz condition, which is the closest frequency to the behaviourally derived FFF threshold surrogate (cf. see the "Methods" section for more details). For the SC ROI, but not for VC or LGN ROIs, the fMRI-derived neurometric function had a sigmoidal shape, which resembled the behavioural psychometric function, with a "chance-level" threshold of 19.8 ± 2.4 Hz.

To investigate the potential involvement of other structures outside the visual pathway with temporal discrimination, several ROIs along different brain regions were drawn, and the corresponding time

profiles were plotted (Fig. S7). We found no clear trend or tracking of stimuli in these signals for any of the areas investigated outside of the visual system, as also reflected by the absence of significant voxels in the maps shown in Fig. 2.

## Electrophysiology reveals that activation/suppression of neural activity underpin fMRI signal transitions in SC

As the SC showed the strongest behaviour–fMRI correlation, we targeted it for electrophysiological recordings (Fig. 3). Fluorescence microscopy images (Fig. 3A) validated the optimal angle of the silicon probe to record from the superficial layers of SC (sSC). Figure 3B and C detail local field potential (LFP) traces (cf. rightmost plot of Figs. 3B and S8 for zoomed-in plots) and total spectral power over time between 1 and 50 Hz for representative stimulation conditions, respectively (cf. Fig. S9 for remaining conditions). At 1 Hz, individual flashes clearly elicited periodic LFP transients and strong power increases. At 15 Hz, sharp power increases are observed at the beginning and end of the stimulation block, while more modest power increases are observed during stimulation in the 15 Hz band (and some of its harmonics), transients shown in more detail in the rightmost plot of Fig. 3B. By contrast, in the 25 Hz condition, periodic transients are

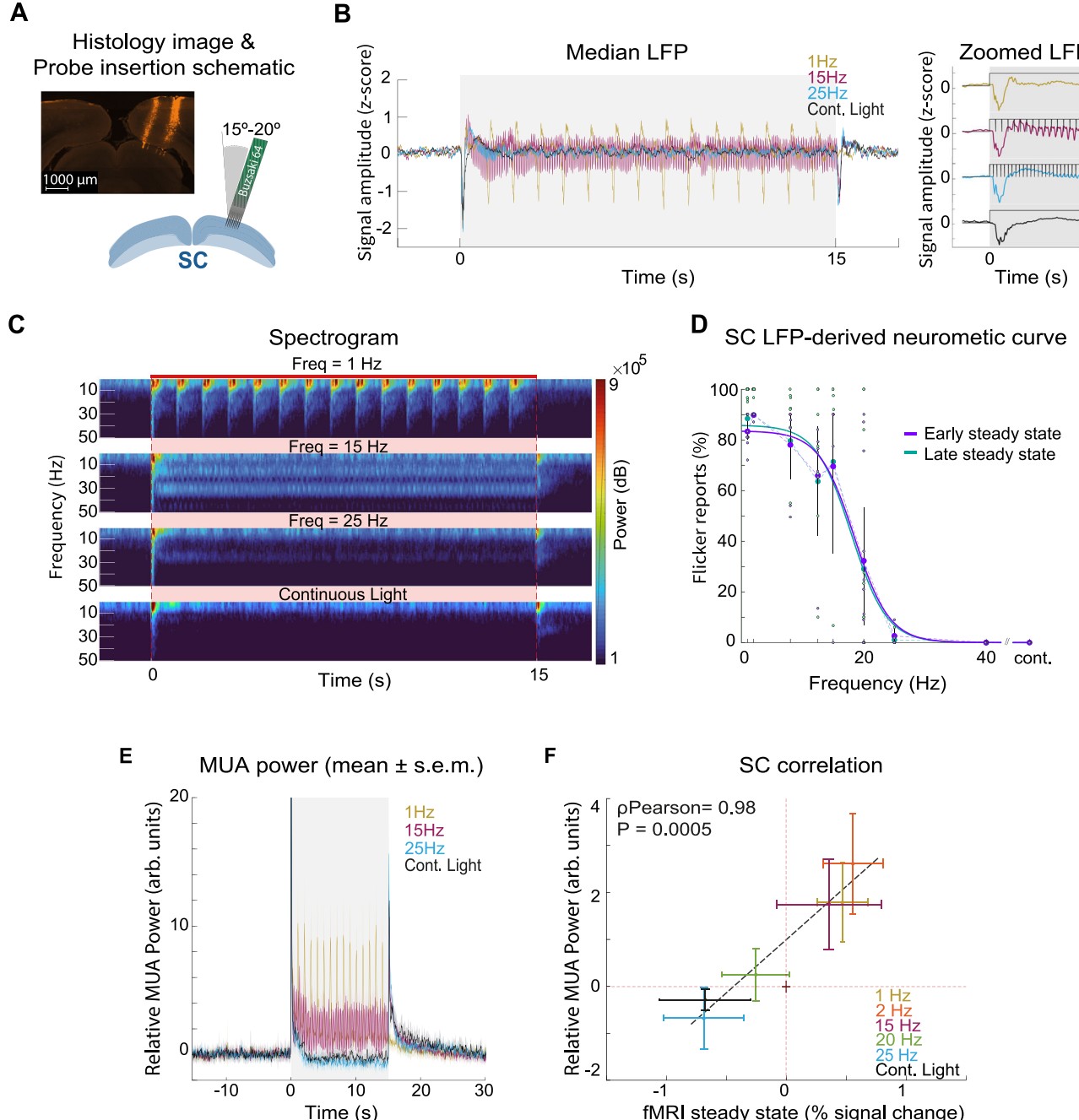

**Fig. 3 | Electrophysiology results (total _n_ = 20, for every individual frequency _n_ ≥ 4). A** Probe insertion schematic and fluorescence microscopy image. **B** Median LFP traces. LFP traces show individual flash-induced LFP oscillations for the 1 Hz condition and onset and offset peaks for the higher frequencies. Zoomed LFP plots show the first second of responses where flash-induced LFP (with increasingly reduced amplitude) can be observed until the 25 Hz stimulation regime. **C** Spectrograms between 1 and 50 Hz for 1, 15, 25 Hz and continuous light stimulation regimes. **D** "Steady-state" LFP-driven neurometric curves. The dots represent mean flicker reports for each animal while the error bars represent the 95% confidence interval. For frequencies 1, 2, 8, 12.5, 15, 20, 25, 40 Hz and continuous light, the used number of trials was 440, 380, 230, 230, 210, 380, 210, 230 and 180, respectively. The fitted sigmoids reveal "chance-level" thresholds of 18.0 ± 1.7 and 17.7 ± 2.4 Hz, for the early and late "steady-state" intervals, respectively. These

values are in accordance with the ones obtained in the SC fMRI-driven neurometric curves and behavioural psychometric curve. **E** Mean ± s.e.m MUA relative power plots across animals. These plots reveal a decreased power during stimulation as the frequency of stimulation increases. Offset signals become evident for the high frequencies. The two highest frequency regimes show power decreases during stimulation below baseline level. **F** Correlation between "steady-state" MUA relative power and fMRI percent signal change. Coloured circles represent the average response of electrophysiological and fMRI sessions and error bars represent the standard deviation across runs. A high correlation coefficient of _R_(4) = 0.98, (_P_ = 0.0005, 95% CI = [0.84,1.00], two-tailed Pearson's correlation) shows a tight relationship between the two measurements. NBRs at high stimulation frequencies correlate with strong MUA power reductions. LFP local field potential, MUA multi-unit activity. Source data are provided as a Source Data file.

much reduced (although still present as it can be seen in the rightmost plot of Fig. 3B) while the sharp onset and offset signals are still observed (Figs. 3B, C and S8).

While individual flash-induced LFP power increases can be observed during the entire 1 Hz stimulation regime, a "steady-state" is reached after ~1-2 s of stimulation for 15 and 25 Hz. To investigate how discriminable the "steady-state" periodicity is in the transients evoked by stimuli of different frequencies in a single trial, we computed LFP-derived neurometric curves for two different intervals beginning at 2 or 8 s after stimulation onset with a 5 s duration, respectively (Fig. 3D). These curves were calculated by computing the fraction of trials in which the integral of the LFP power spectrum at the frequency of the stimulus used in that trial, was higher than the mean (across all the 20 Hz stimuli) of the integral of the LFP power spectrum at 20 Hz (cf. see the "Methods" section for more details). Both "steady-state" intervals revealed similar LFP-derived neurometric curves with chance-level thresholds similar to those obtained from the behavioural data and fMRI: $18.0 \pm 1.7$ and $17.7 \pm 2.4$ Hz for early and late "steady-state" intervals, respectively.

When the multi-unit activity (MUA) power along the different stimulating frequencies was computed (Fig. 3E), similar trends were observed. Interestingly, the "steady-state" for the two higher frequency conditions (25 Hz and continuous light) drops below baseline levels suggesting inhibition, contrasting with the LFPs at this regime (Fig. S10). To better explain the PBR to NBR transitions observed with fMRI in SC, we tested a broader range of visual stimulation frequencies (Fig. S10). Higher stimulation frequencies clearly produced reductions in MUA power—suppression of neural activity—alongside larger NBRs.

Figure 3F investigates the relationships between fMRI time profiles and their MUA counterparts by showing the correlation between the mean "steady-state" signals of the two modalities (a similar analysis is shown for the LFP band in Fig. S11). The two signals are highly correlated (also agreeing with recent work[38]) and reveal how multiphasic BOLD responses appear to be better characterized by higher frequency band signals such as the MUA ($\rho_{Pearson}(4) = 0.98$ ($P = 0.0005$, 95% CI = [0.84,1.00], two-tailed Pearson's Correlation)). Stimuli below the chance-level threshold clearly induced fMRI and MUA positive signals, while stimuli above this threshold led to a strong reduction of fMRI and MUA signals.

The fMRI-BOLD signals are naturally delayed compared with their fast MUA counterparts due to the complex neurovascular coupling mechanisms[39–43]. Figure S12 shows the 1 and 25 Hz LFP and MUA curves convolved with an hemodynamic response function (HRF). Interestingly, the convolved electrophysiological signals predict well the timing observed for the onset/offset fMRI signals timing (fast fMRI curves were acquired for this figure with a temporal resolution of 500 ms).

### V1 feedback acts as gain control for activation to suppression transitions in SC

Since the SC receives cortical feedback from the primary visual cortex (V1) (Fig. 4A), we sought to investigate whether the transitions from PBR to NBR responses strongly depended on such feedback connections. To this end, V1 was bilaterally lesioned via ibotenic acid in 13 animals. Ex-vivo histological images and in-vivo structural MRI scans (Fig. 4A, bottom row) confirmed the localization of the lesions in V1. Interestingly, one-week post-lesion, secondary cortical regions showed enhanced PBRs when stimuli were delivered at 1 Hz, while at higher frequencies, the cortical NBRs observed in the sham were not apparent in the lesion group. Strikingly, the SC PBRs and NBRs appeared attenuated in the lesioned animals when compared to the sham group (Fig. 4B). Time profiles comparing the two groups for three representative frequencies are shown in Fig. 4C for SC signals (cortical and thalamic responses in the lesion group are shown in Fig. S13). Clearly, animals with V1 lesions exhibited stronger onset and offset peaks as well as weaker negative "steady-state" fMRI signals in

the SC. SC signals at lower frequencies were also affected by V1 lesions (Fig. 4C), therefore suggesting that V1 exerted a gain control effect in SC, but was not necessary for producing suppression of activity in SC at higher flicker frequency. Note that the calculated SNR for the control group in the SC and LGN was $13.3 \pm 6.1$ and $9.6 \pm 2.6$, respectively, while the calculated SNR for the (V1) lesioned group in the SC and LGN was $13.0 \pm 3.5$ and $10.8 \pm 2.9$, respectively—suggesting that potential SNR differences due to V1 lesions did not confound this analysis.

To confirm that the SC could still follow individual flashes in the V1 lesioned-animals as it did in the control conditions, electro-physiological responses were recorded in 6 animals with lesions in V1 (Fig. 4D). The trends strongly resembled those of the control group, with individual-flash evoked responses still visible at 25 Hz. MUA power plots (Fig. 4D rightmost panel), also exhibited reductions in power during the stimulation period for higher stimulation frequencies as in control conditions (Fig. 3E). In addition, an MUA power reduction below baseline can be observed already at 25 Hz stimulation, as was observed for controls.

Figure 4E proposes a hypothesized mechanism underpinning these SC signals. Within the SC itself, we consider there are two overlapping effects: (i) a "novelty detection" effect (onset/offset signals) and (ii) "frequency discrimination" perception—a constant effect of activation/suppression of SC activity modulated by stimulation frequency. When the SC can still follow individual flashes (at low frequencies), each flash represents a novelty event and therefore single flash-induced activations are observed. On the other hand, when SC neurons are no longer capable of regaining their full excitability in-between light flashes (for high frequencies when the inter-stimulus interval is reduced), the stimulation period becomes an illusory continuous light with no novel events occurring while the light is "on". In the latter scenario the novelty events only occur when the stimulus starts and once again when the stimulus ends, corresponding to the two peaked signals measured for higher frequencies. In addition, a cortical gain control also takes place and modulates the SC responses—when frequencies are low it potentiates the evoked responses, and when frequencies are high, it increases the suppression in the SC to avoid instigation of the novelty detection. Other contributions within the visual pathway cannot, of course, be ruled out in participating in this interaction.

## Discussion

Investigating the brain's ability to "fuse" sensory inputs and induce complex mechanisms such as the continuity illusion (an analogue of which can also occur in the auditory domain[44]) is critical for better understanding phenomena such as visual perception and its encoding along the visual pathway.

Studies focusing on the light entry point, the retina[18–20], in species that are more reliant on vision such as monkeys[30,45], dogs[32], cats[21,33,46,47] or birds[19,24,48], initially proposed that FFF thresholds were solely dependent on retinal function and rod/cone composition[22,49,50]. However, behaviourally-derived FFF thresholds were always found to be lower than those derived from retinal electrophysiological recordings[19,24,35,36]. Hence, temporal resolution cannot be limited by the retina's ability to resolve flickers but rather reflects processing downstream in the visual pathway, where thresholds are likely modified at various stages[19,28]. Attempts to measure correlates of the visual continuity illusion at the higher order perception levels were carried out at the last neural information processing stage (VC)[21–23], where the lowest FFF thresholds were reported in several species[51–53]. Thresholds for individual cortical cells varied across a broad range of frequencies and importantly, cortical FFF thresholds were lower than behaviourally-observed thresholds[22]. Hence, we hypothesized that the area most relevant for switching between dynamic and static vision modes must be located elsewhere, and applied a multi-level investigation to find it.

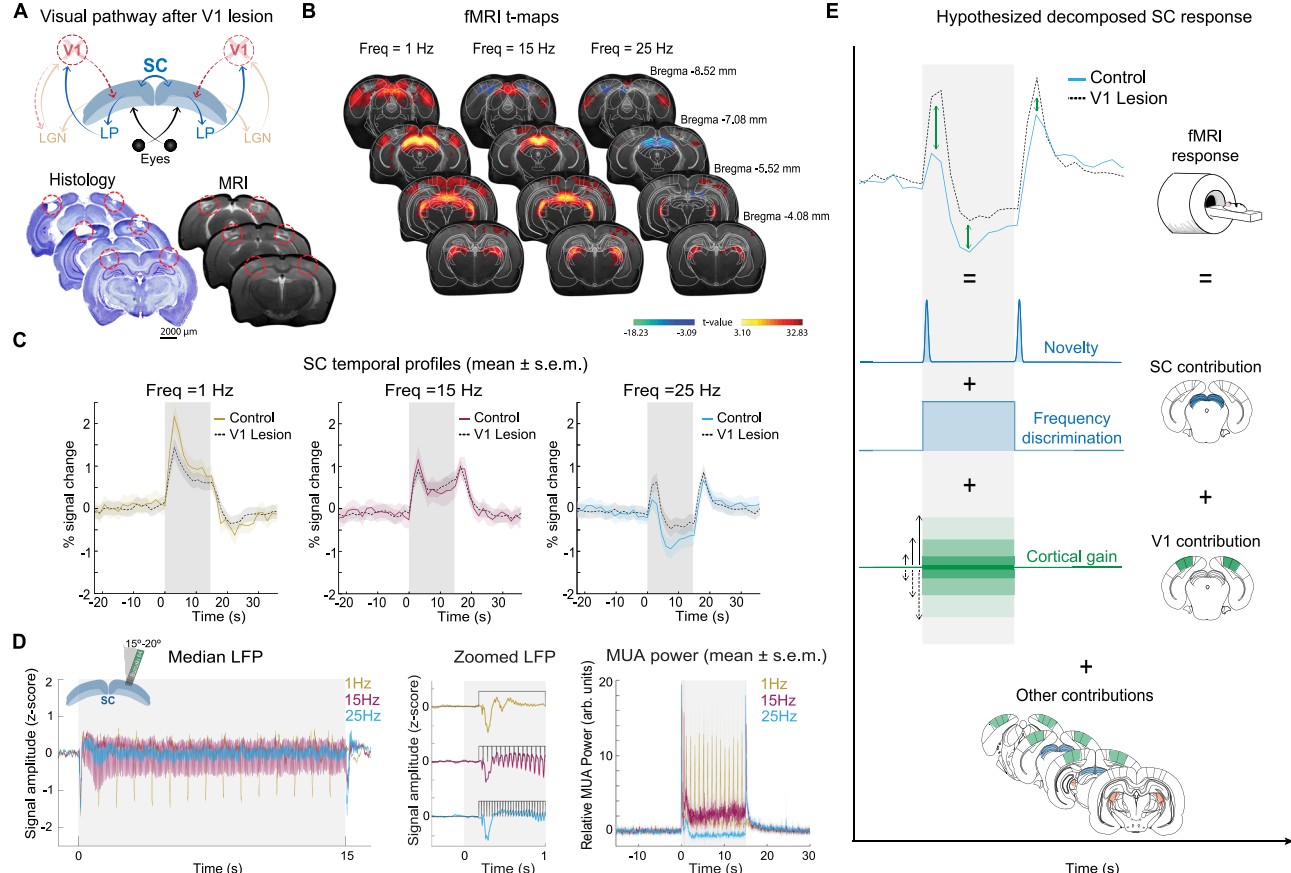

**Fig. 4 | fMRI ibotenic acid lesions results. A** Schematic of the visual pathway after V1 lesion highlighting reduced cortical feedback (TOP); histological and structural MR images (BOTTOM) confirming the lack of brain tissue in the lesion site. **B** fMRI t-maps for representative frequencies ($n = 13$). As the frequency of stimulation increases, transitions from PBRs to NBRs appear in SC but not in VC. Secondary visual cortical areas show enhanced PBRs at 1 Hz compared to the control regime, probably reflecting plasticity events that took place between the lesion induction and the fMRI experiments. **C** Mean ± s.e.m. SC temporal fMRI profiles across animals ($n = 13$). SC temporal profiles reveal marked positive to negative BOLD shifts and the onset and offset signals remain present after the lesion. Reduced amplitude of both positive and negative SC BOLD responses highlights the gain effect from V1 feedback projections. **D** Electrophysiology results. Electrophysiological signals were recorded in lesioned animals ($n = 6$) to confirm the presence of the SC oscillatory responses. From the left and middle plots, single flash-evoked responses are observed until 25 Hz. Onset and offset signals are still observable with the latter only present for the higher stimulation conditions. MUA power plots show a power reduction with stimulation frequency. **E** Hypothesized decomposed SC response. A hypothesized mechanism is proposed based on the fMRI results. The SC response has at least two different contribution sources: a contribution within SC (novelty and a constant frequency discrimination perception) and a cortical contribution acting as gain control. Possible contributions from other structures within the visual pathway cannot be ruled out. V1 primary visual cortex, PBR positive BOLD responses, BOLD blood oxygenated level dependent, SC superior colliculus, VC visual cortex, MUA multi-unit activity. Source data are provided as a Source Data file.

Our behavioural task was designed to avoid biasing animals towards one port and the reward system was carefully chosen to motivate animals to report their perceptions about stimulus continuity (c.f. Behavioural task supplementary discussion for a more detailed discussion on the task's contingencies). The FFF threshold surrogate measured here ($18 \pm 2$ Hz, Fig. 1C) agrees well with prior literature (~21 Hz in a different task[22,54]). Interestingly, the whole-pathway fMRI approach revealed that signals in SC—but not VC or LGN signals—bear a close relationship with the behavioural results: flicker reports above chance-level and associated PBRs indicated an activation of the SC, while below chance-level, strong NBRs were observed during the bulk of the stimulation period (apart from the onset/offset signals, vide-infra), which intensified as the stimulation frequency was increased. Subsequent electrophysiological recordings in SC lent further credence to the fMRI findings and revealed that suppression of MUA is responsible for the strong NBR responses.

A plausible mechanistic hypothesis for these signals in the SC would suggest activation upon perceiving flickering light (PBRs) and suppression of neural activity (NBRs) at high frequencies where the

light "fusion" mechanisms take place and the animals report to have entered the dynamic vision mode. Taken together, our findings suggest that activation/suppression balances in SC activity are important contributors for temporal frequency discrimination and potentially for entering the continuity illusion regime.

Anatomically, two visual sub-pathways co-exist: the extra-geniculate pathway (including SC) and the geniculate pathway (including LGN). Experiments lesioning either one or the other revealed different roles in FFF threshold determination[21,33]: while the geniculate pathway mediates high flicker frequencies, the extra-geniculate pathway mediates lower frequencies. The SC would behave as a lowpass filter limiting high-frequency perception, in-line with our findings that the SC plays an active role in temporal frequency discrimination. Experiments looking into the involvement of the VC in flicker discrimination revealed that while humans became insensitive to any form of light stimulation following cortical ablation, lower-order species such as cats[55] and albino rats[51] were still able to discriminate flicker from steady light. This suggests subcortical "fusion" mechanisms in lower-order species. In this study, lesioning V1 revealed that

this region is not necessary for generating the observed SC onset and offset peaks as well as the transition from activation to suppression, but rather, it exerts a gain effect likely arising from cortical feedback into SC. In fact, in several animal species[56–59] cortical and retinal inputs in the SC are aligned, and an instructive role of retinal inputs in the mapping and alignment of cortical visual projections to the SC was suggested (probably through spontaneous activity prior to eye-opening). In line with the map alignment mechanisms suggested above and the cortical gain effect hypothesized from the V1 lesion experiment, superficial SC neurons have been shown to inherit basic feature selectivity from the retina and modulate their response magnitude depending on cortical input[60]. This agrees well with our conclusion that SC plays a major role in temporal frequency discrimination while also highlighting the involvement of other structures along the visual pathway.

While only a few studies have investigated the SC in the context of the continuity illusion[21,33], it has been much more widely studied in the context of response habituation[61,62] (RH), a phenomenon which, similarly to the continuity illusion effect, occurs at high stimulation frequencies. RH is expected to serve as a form of short-term memory for familiar versus novel information based on the dynamic adjustment of response thresholds. Studies[62–65] have suggested a mechanism of feedback inhibition to underpin RH: the co-activation of excitatory and inhibitory neurons leads to a long-lasting inhibition blocking responses to subsequent stimulus presentation at high enough frequencies. The RH effect could be a contributor to the continuity illusion effect (probably among other mechanisms occurring along the visual pathway) and, if indeed the two phenomena are related, the continuity illusion effect in the SC would result, in part, from inhibitory processes –in-line with the measured neuronal suppression at high stimulation frequencies both via fMRI and MUAs.

This study not only confirms the ability of BOLD-fMRI to resolve onset/offset signals previously reported in VC and SC at high-frequency stimulation regimes[38,66] but also sheds light on the possible behavioural relevance of such signals. Onset peaks have previously been described with electrophysiology in the SC[67–69]–at different stimulation frequencies, responses to the first flash remained similar and only subsequent responses appeared reduced in amplitude. Regarding the offset peak, the co-activation of excitatory and inhibitory neurons[65] alone is unlikely to fully explain these signals, suggesting that the measured neuronal suppression occurring during the stimulation period likely reflects a more active process, and not solely the result of different lasting simultaneous effects or adaptation. One interpretation could be that when individual flashes are no longer perceptible, the entire stimulation period is "fused" and integrated as "one long flash" and the SC onset/offset peaks reflect brightness changes or novelty detections at the edges of stimulation–from dark to bright (onset peak) and from bright to dark (offset peak).

Our work also provides important insight into the ongoing debate on the nature of negative BOLD signals and their underlying biological underpinnings[70–72]. The correlations between MUA and NBRs in this study strongly point to neuronal supression[44,72–80] as the most probable scenario, and provide a system where the amplitude of such signals can be modulated by a simple experimental variable–the stimulation frequency–which could serve as an experimental handle for future research into the mechanisms underlying the NBR neurovascular coupling.

Finally, as in every study, several limitations can be noted. First, the sedation state of the animals differed between behaviour (awake) and recordings during fMRI and electrophysiology data acquisition (where animals were lightly sedated with medetomidine, an α2 agonist[81]). This could potentially account for the very slight differences in perceptual behavioural reports and neural/BOLD signal transitions, although these differences were indeed small. The effects of medetomidine sedation were also shown to be small when comparing the

baseline neural activity to the awake state[82] and a (qualitative) comparison with visual responses in awake mice[83] shows similarity to those observed in the present work. Furthermore, studies performed in rats using the same dosage of medetomidine as in this work have reported functional connectivity that was partially attributed to wakefulness[84]. In addition to the good agreement between the measurements from different modalities shown in this study, this suggests that the sedated state induced by medetomidine did not play a critical effect in our observations.

A second potential confounder is that while fMRI and electrophysiology measurements were performed on mixed cohorts of males and females, the behavioural experiments were performed solely on female rats due to the size constraints of the behavioural setups. However, given the agreement between behaviour, fMRI, and electrophysiology, and the low-level nature of these behavioural experiments, we do not expect this sex effect to be large. Another difference between behavioural measurements and fMRI/electrophysiology is the different levels of oxygen concentration. Our behavioural experiments were performed under normoxia–21–23% $O_2$ - while the fMRI and the electrophysiological experiments were performed under hyperoxia– 95% $O_2$ -, to enhance fMRI CNR (Fig. S3), particularly in the negative BOLD regime. However, this high oxygen concentration regime is not expected to induce significant changes at the neural activity level[85,86], in-line with the similar shape and timings of fMRI responses under different oxygen concentrations shown in Fig. S3. Most studies examining the effect of hyperoxia on behavioural responses use different kinds of physically demanding tasks[87,88] which are irrelevant for our low-level task. While we cannot completely exclude that oxygen concentration accounted for some subtle effects, the agreement across modalities again suggests that these confounds were minor.

The dark adaptation period was also slightly different between modalities. Previous work in Long-Evans rats suggests that complete dark adaptation is achieved after about 30 min[89]. In our fMRI and electrophysiology groups, we allowed for a minimum of 45 min of dark adaptation before data was acquired. However, in the behavioural task, the animals were inside the box for approximately 15 min before the beginning of the trials, which could potentially confound our results. To test for this effect, we reanalysed the behavioural data, but now considering only trials that occurred from the point in which 30 min had already elapsed inside the dark box (shown in Fig. S2). Results yielded indistinguishable curves when compared to the full dataset (i.e. with only 15 min of light adaptation). The similarity between psychometrics could suggest that the animals started to be more engaged in the task (performing more trials) later on in the task (e.g. after 30 min inside the box) or that our animals sufficiently adapted to dark conditions already after 15 min for the purpose of this particular task. Therefore, we conclude that adaptation differences due to the slightly shorter time in the behavioural experiments could not have caused any significant bias.

A further difference between the conditions involves the stimulus duration: Animals made their behavioural reports after 1 s of stimulus presentation, while in the fMRI and electrophysiological sessions, stimuli lasted for 15 s. Our "steady-state" analyses (Fig. 3D) were designed to take these differences into account and indeed revealed that this is not a major confounding effect since, although the report takes place early on, the stimulus perception is maintained throughout the stimulation period. Furthermore, as shown in Fig. S12, fMRI signals track MUA signals: the timing of NBRs onset/offset and steady state at 1 and 25 Hz could be fully predicted from the MUA curves convolution with a conventional HRF–a good indication that "steady-state" fMRI signals can be used with good confidence as a proxy for the MUA recordings and consequently, also explain the behavioural reports.

In conclusion, we investigated the shift between static and dynamic vision modes–related to the visual continuity illusion effect– using a powerful multimodal approach encompassing behaviour,

whole-pathway fMRI, electrophysiological recordings in the SC, and cortical lesions, aiming to bridge the disparity between behaviourally-reported and retinal/cortical FFF thresholds. We find that FFF threshold proxies in the SC, but not VC or LGN, agree with FFF surrogates from behaviour and electrophysiology. We have shown that in the SC, neural activity is highly suppressed above the measured FFF threshold and therefore we conclude that activation/suppression balances in the SC play a crucial role in encoding the transition from low to high frequency discrimination. Finally, lesions in V1 highlighted a cortical gain effect component and supported our hypothesis that the transitions from activation to suppression regimes are intrinsic to the SC. Our work underscores the potential of rodent fMRI in unraveling mechanistic insights at the pathway and population levels and its relevance for future investigations of dynamic processes such as disease or plasticity[90].

## Methods

All animal care and experimental procedures were carried out according to the European Directive 2010/63 and pre-approved by the competent authorities, namely, the Champalimaud Animal Welfare Body and the Portuguese Direcção-Geral de Alimentação e Veterinária (DGAV).

In this study, $n = 62$ adult Long-Evans rats were used with *ad libitum* access to food and water (except for the behavioural animals which were water-deprived) and under a normal 12 h/12 h light/dark cycle. The fMRI control group consisted of 11 female and 7 male rats and animals were $17.1 \pm 8.3$ weeks old and weighed $351.4 \pm 75.6$ g on average. The fMRI V1 lesioned group consisted of 10 female and 3 male rats and animals were $16.4 \pm 3.7$ weeks old and weighed $375.9 \pm 75.1$ g on average. The fast fMRI group consisted of 5 female and 1 male rats and animals were $21.0 \pm 8.0$ weeks old and weighed $332.8 \pm 53.0$ g on average. The electrophysiology control group consisted of 14 female and 6 male rats and animals were $19.0 \pm 7.0$ weeks old and weighed $405.9 \pm 70.2$ g on average. The electrophysiology V1 lesioned group consisted of 3 female and 3 male rats and animals were $19.0 \pm 4.1$ weeks old and weighed $463.0 \pm 106.6$ g on average. This study used both males and females without any specific criteria as this variable is not expected to affect results.

In all three modalities used in this present work, animals were fully naïve to the presented stimuli prior to the procedure. This was done in order to accomplish a fair comparison between groups and to probe the natural FFF threshold surrogate of the animals. For this reason, and to not influence the measured thresholds, animals (1) did not perform any behavioural task prior to MRI or electrophysiological experiments and (2) were trained in the behavioural task for a relatively short period of time, in an attempt to prevent behavioural strategies not directly related to their perception of the light, from skewing the results.

Data analysis was performed using MATLAB (The Mathworks Inc., Natick, MA, USA, v2016a and v2018b).

### Animal preparation

Animals were water-deprived for 3 days prior to the behavioural task and their weights were monitored during the entire duration of the study. Weight loss was capped at 10% body weight while water-deprived.

Before the MRI and electrophysiological acquisitions, animals were anaesthetized with 5% isoflurane (Vetflurane, Virbac, France) for 2 min and were then moved to either the MRI bed or the stereotaxic set-up, respectively. Animals were later sedated with a medetomidine solution (1:10 dilution in saline of 1 mg/ml medetomidine solution—Vetpharma Animal Health, S.L., Barcelona, Spain) by injecting a subcutaneous bolus (bolus = 0.05 mg/kg) while isoflurane dosage kept on being reduced. During fMRI experiments the medetomidine bolus was administered 5 min after the initial induction while during the electrophysiological recordings, the bolus was administered after the

surgery was done (~40 min). Ten minutes after the bolus was administered, a medetomidine constant infusion of 0.1 mg/kg/h, delivered via a syringe pump (GenieTouch, Kent Scientific, Torrington, CT, USA), started. Isoflurane was gradually reduced for the next 10 min until it was stopped and animals subsequently remained only under medetomidine sedation throughout the entire MRI and electrophysiological data acquisition. To achieve efficient isoflurane washout, acquisitions were always started between 50–60 min after bolus injection. After the experiments, sedation was reverted by injecting the same amount of the initial bolus of a 5 mg/ml solution of atipamezole hydrochloride (Vetpharma Animal Health, S.L., Barcelona, Spain) diluted 1:10 in saline. At the end of the electrophysiological recordings, the animal was euthanized with a solution of sodium pentobarbital (delivered via intraperitoneal injection) and the brain was extracted and maintained in 4% PFA for a period of 12–24 h before slicing for further microscopy imaging.

### Behaviour

To investigate the biological underpinnings of temporal frequency discrimination, we measured FFF threshold surrogates behaviourally.

Water-deprived animals ($n = 7$, 14–20 sessions) were placed in a box with three ports and one blue LED placed on the ceiling in order to illuminate in a homogeneous way the entire behavioural setup (therefore the entire field of view of the animals).

Rats were trained to associate each lateral port to either continuous or flicker light (Fig. 1A). The central port was used for trial initiation and the animal had to poke in for ~200 ms. After this time, the visual stimulus started: a blue LED on the top of the behaviour box turned on displaying either a continuous or a flickering light at different frequencies (keeping the pulse width of 10 ms while modulating the ISIs). The animal had to wait 1000 ms until a pure tone was played (with a frequency of 10 kHz), after which it could report its choice. The light would turn off only after the animal left the central port. The animals were trained to poke on one side for continuous light, and on the other side for flickering light (the contingency was counterbalanced between animals). If the animal selected the correct port a water reward of 25 µl was delivered, otherwise a time penalty of 5000 ms initiated along with a burst of white noise to signal the mistake (Fig. 1B).

During the training phase, animals were presented with either true continuous light or a flicker frequency of 2 Hz. The initial fixation time and waiting period were set to 10 and 0 ms, respectively, and were incremented by 1 ms each time the animal completed a trial. After reaching a performance >80% with these two easy conditions, other frequencies were introduced and presented in a random manner.

During the task, 50% of trials consisted of continuous light while the other 50% consisted of a flickering light at different frequencies: 1, 2, 8, 12.5, 15, 16, 18, 20, 25, 40, and 50 Hz. This was done to prevent one of the pokes from delivering more rewards. Within the 50% of flicker trials, only 10% contained the higher frequencies (above 8 Hz): the so-called "probe trials" where the animal might enter the continuity illusion regime. These "probe trials" were rewarded in the flicker port. A supplementary discussion is presented in Fig. S1.

The training of the animals in the general contingency of the task—where only two frequencies (2 Hz and continuous light) were presented—took on average $3.0 \pm 1.0$ weeks while the assessment of their FFF threshold surrogate—where more frequencies, including probe frequencies, were presented—took place over $3.8 \pm 0.4$ weeks. The behaviour data was acquired using a custom MATLAB code (with psychotoolbox-3).

### MRI acquisitions

Functional magnetic resonance imaging (fMRI) data was acquired to observe blood-oxygenation-level-dependent (BOLD) contrast modulations induced by the different frequencies along the entire rat visual

pathway (Fig. 2). The experiments were conducted using a 9.4 T Bruker Biospec MRI scanner (Bruker, Karlsruhe, Germany) equipped with an AVANCE III HD console including a gradient unit capable of producing pulsed field gradients of up to 660 mT/m isotropically with a 120 μs rise time was used. An 86 mm quadrature coil was used for radio frequency transmission and a 4-element array cryoprobe[91] (Bruker, Fallanden, Switzerland) was used for signal reception. The software running on this scanner was ParaVision® 6.0.1. During the entire duration of the fMRI experiments, animals breathed a mixture of 95% oxygen and 5% medical air. The animal's temperature was monitored during the entire experiment ($36.5 \pm 1.0$ °C) with an optic fibre rectal temperature probe (SA Instruments, Inc., Stony Brook, New York, USA) and was regulated using a heating system consisting of circulating water. Respiratory rate was measured using a pillow sensor (SA Instruments Inc., Stony Brook, USA).

For correct slice placement along the visual pathway, an anatomical T$_2$-weighted Rapid Acquisition with Refocused Echoes (RARE) sequence was used (TR/TE = 1600/36 ms, RARE factor = 8, echo spacing = 9 ms; averages = 3; FOV = $18 \times 16$ mm$^2$, in-plane resolution = $168 \times 150$ μm$^2$, slice thickness = 800 μm, $t_{acq}$ = 1 min 3 s). Several control experiments were performed to determine the best conditions for data acquisition (Figs. S3–S5). The functional MR imaging was acquired using a Spin-Echo Echo-Planar Imaging (SE-EPI) sequence (TE/TR = 40/1500 ms, PFT = 1.5, FOV = $18 \times 16.1$ mm$^2$, in-plane resolution = $269 \times 268$ μm$^2$, slice thickness = 1.5 mm, $t_{acq}$ = 6 min 50 s).

### Electrophysiological recordings

To understand the neural underpinnings of temporal frequency discrimination, MRI-targeted electrophysiological recordings were performed in the right superior colliculus (SC) inside a double-wall anechoic chamber, allowing for outside noise and light contamination reduction and acting as a Faraday cage.

Each recording lasted ~5 h and both stimulation and sedation conditions were kept as similar to the MRI acquisitions as possible.

During surgery, the animal was kept under the effect of 3% isoflurane and the eyes were protected from light by covering them with an opaque gel (Bepanthen augen und nasensalbe). To avoid electrical noise, the animal's temperature was kept within physiological parameters with the aid of an air-activated heating pad.

Following stereotaxic coordinates (using the 6$^{th}$ Edition of Paxinos & Franklin's rat brain atlas[92]) calculated from previous anatomical T$_2$-weighted RARE MRI scans, a $2 \times 3$ mm craniotomy was opened in the skull above the right SC; and the dura was removed under a stereoscope using a small needle (BD MicrolanceTM 518 $0.3 \times 13$ mm).

Extracellular recordings were performed using a 64-channel (8 shanks × 8 sites) silicon probe (Buzsaki64, NeuroNexus, Ann Arbor, MI). The probe was connected to an analogue 64-channel headstage (Intan) and positioned with a manual micromanipulator at an angle with the vertical between 15° and 20°, for better alignment with the SC and in order to avoid blood vessels. Prior to brain insertion, the probe was stained with DiI (Vybrant™ DiI Cell-Labelling Solution) for post-hoc confirmation of the recording location with microscopy imaging (Fig. 3A). The skull cavity was kept moist with saline during this entire process.

The data was digitized at 16 bit and stored for posterior offline processing using an Open Ephys acquisition board (Open Ephys v0.5.5.1 to v0.6.4) at a 30 kHz sampling rate.

### Visual set-up and paradigm for fMRI and electrophysiological acquisitions

The eyes of the animal were hydrated during the acquisition with an ophthalmic gel (Visidic gel Bausch + Lomb) and a bifurcated optic fibre connected to a blue LED ($\lambda$ = 470 nm and I = $8.1 \times 10^{-1}$ W/m$^2$) was placed horizontally in front of each eye of the animal (spaced up to 1 cm) for binocular visual stimulation (Fig. 2A). In the MRI the LED light is

reflected inside the bore while in the electrophysiology the rat's head was covered with reflective material to ensure that the blue light stimulated the entire field of view of the animal.

For the MRI acquisitions, the blue LED was connected to an Arduino MEGA260 receiving triggers from the MRI scanner and used to generate square pulses of light. For the electrophysiological acquisitions, the blue LED was connected to an I/O board built in-house and controlled through MATLAB 2016a.

Several stimulation frequencies were used–0.25, 1, 2, 15, 20, 25 Hz and continuous light. For the flickering stimuli, the flash duration was kept constant at 10 ms in order to modulate inter-stimulus intervals (ISIs).

The stimulation paradigm consisted of a 15 s stimulation period interleaved with 45 s rest periods (Fig. 2A). Both the fMRI and electrophysiological acquisitions consisted of an initial resting period of 45 s followed by 6 and 10 cycles of the experimental paradigm, respectively.

### Ibotenic acid lesions

To investigate the effect of cortical feedback projections in SC signal modulations, animals ($n$ = 13) were injected bilaterally with an ibotenic acid[93] solution (excitotoxic agent, 1 mg/100 μL) in the primary visual cortex (V1). The acid was injected using a Nanojet II (Drummond Scientific Company). Animals were anesthetized with isoflurane (anaesthesia induced at 5% concentration and maintenance below 3%), and a scalpel incision was made along the midline of the skull, the skin retracted and the soft tissue cleaned from the skull with a blunt tool.

Coordinates for the craniotomies and amounts required for the injections were determined for each individual animal based on T$_2$-weighted anatomical images acquired before the surgery. We injected in 5 different AP coordinates; two injection pulses on the one injection site for the first AP coordinate; and 4 injection pulses for each of the following injection sites, two per AP coordinate, for full coverage of V1. Each injection pulse was administered at a rate of 23 nL/s and 2–3 s between pulses and each pulse consisted of 32 nL. Waiting time before removing the injection pipette after the last pulse was 10 min. The craniotomies were then covered with Kwik-Cast™ and the scalp was sutured.

### Data analysis

Data analysis for all modalities was done with custom codes running in MATLAB (The MathWorks Inc., Natick, MA, USA).

**Behaviour analysis.** We analysed the behavioural task as a detection task[94,95], in which the rat had to detect whether a flashing stimulus or continuous light was presented. The percentage of responses to the flicker light port was calculated and, to estimate the FFF threshold surrogate, a sigmoid function was fitted to the averaged animal response and the intercept at 0.5 ("chance level") was taken. For the standard deviation calculation, performances were separated by individual sessions and a bootstrap method was applied by resampling with replacement (50 iterations). For the Pearson correlations with fMRI results the average performance of all sessions was computed and the standard deviation across sessions is shown in the error bars.

**fMRI analysis.** For the MRI data, a general linear model (GLM) analysis was conducted along with a region of interest (ROI) analysis to investigate the temporal dynamics of activation profiles.

**GLM analysis.** Pre-processing steps included manual outlier removal (a spline interpolation was made taking the entire time course), slice-timing correction (using a sinc-interpolation) followed by head motion correction (using mutual information). Data was afterwards co-registered to the T$_2$-weighted anatomical images, normalized to a

reference animal and smoothed using a 3D Gaussian isotropic kernel with a full-width half-maximum corresponding to 1 voxel (0.268 mm).

A fixed-effects group analysis was performed. The stimulation paradigm was convolved with an hemodynamic response function (HRF) peaking at 1 s. A one-tailed voxelwise *t*-test was performed, tested for a minimum significance level of 0.001 with a minimum cluster size of 20 voxels and corrected for multiple comparisons using a cluster false discovery rate test (FDR).

**ROI analysis.** For the ROI analysis, the 6[th] Edition of Paxinos and Franklin's rat brain atlas[92] served as guidance for the manual ROI delineation (Fig. 2B). The individual time profiles were detrended with a 5[th]-degree polynomial fit to the resting periods in order to remove low-frequency trends and were then converted into percent signal change relative to baseline. For each run, the six individual cycles were separated and averaged across all animals to obtain the averaged response within each ROI (along with the standard error of the mean).

For the "steady-state" time portion calculation the averaged fMRI response per stimulation frequency for each ROI was resampled to a 4 Hz sampling frequency. A comparison was then performed between two consecutive sliding windows comprised of 4 time points (sharing one central time point). The "steady-state" was assumed to be reached when the difference between medians of the two windows was <0.04. The "steady-state" percent signal change for each run for different animals was then averaged and the averaged baseline value was subtracted in order to correct for any remaining baseline fluctuations. The mean fMRI percent signal change during "steady-state" across animals was then correlated with the mean percentage of reports to the flicker port (Fig. 2D) and with the mean MUA relative "steady-state" powers (Fig. 3F). The correlations were computed using the mean values of "steady-state" for each run/session across animals for each condition (i.e. stimulation frequency) and the error bars represent the standard deviation of these values.

fMRI-driven neurometric curves were calculated using the "steady-state" portion of fMRI signals from individual trials in order to quantify the single-trial discriminability of the fMRI responses as a function of flicker frequency. Outlier trials were removed by using a threshold-based method: trials with values above/below 3 standard deviations from the mean value for that specific stimulation frequency were excluded. A threshold close to the behaviourally derived FFF surrogate (the closest tested frequency was 20 Hz) was chosen. For each trial (cycle) we estimated the proportion of fMRI responses (averaged across animals) which were above the mean value for the 20 Hz condition. A sigmoid function was fitted to the results (as done for the behavioural data) and the "chance level" threshold (FFF threshold proxy) was considered to be 0.5 of the fit as in the behaviour experiments.

The SNR of the control and lesioned fMRI groups was calculated to investigate the effect of the lesions in the ROI quantifications. For each animal, one functional scan was used for the SNR estimation in the SC and lateral geniculate nucleus of the thalamus (LGN). Two ROIs were drawn: one in the structure of interest and another in a noise region containing a similar number of voxels. The SNR was then calculated by dividing the mean signal in the SC/LGN by the mean noise signal. The calculated SNR for the control group in the SC was $13.3 \pm 6.1$ and in the LGN was $9.6 \pm 2.6$ while the calculated SNR for the V1 lesioned group in the SC was $13.0 \pm 3.5$ and in the LGN was $10.8 \pm 2.9$.

**Electrophysiology analysis.** Electrophysiological data was band-passed with a notch filter at 50, 100 and 150 Hz to remove power line noise and detrended with a linear fit to the entire run.

For each individual run, time-courses were divided into individual cycles and a weighted average across channels was performed in order to obtain the averaged channel cycles for each run. The weighting was based on the integral during the stimulation period of the absolute signal for the lowest stimulation frequency of the group.

The power spectral density (PSD) was calculated using Welch's estimate. A hamming sliding window with 25% overlap was used; the final temporal resolution was set to 50 ms. We further filtered signals in the local field potential (LFP) frequency band (1–150 Hz) and computed the averaged signal for all animals for the different stimulation frequencies (Figs. 3B and S8).

LFP-derived neurometric curves (averaged across rats) were computed to investigate how discriminable was the "steady-state" periodicity in the transients evoked by stimuli of different frequencies in single trials. For the calculation of such curves, the power spectrum of each normalized LFP "steady-state" window signal per trial was computed. We further subtracted to these power spectra the averaged run power spectrum corresponding to a non-stimulation resting period (with the same duration) in order to remove contributions that did not come solely from the stimulation itself. As a metric of comparison for each trial, we used the integral around the stimulation frequency used in that trial and computed the fraction of trials in which this metric was higher than the mean, across all 20 Hz trials, of the integral of the LFP power spectrum at 20 Hz. A sigmoid function was then fitted to the results and the "chance level" threshold was calculated (0.5 of the fit). We computed the LFP-derived neurometric curves for two different time intervals: early and late "steady-state" intervals defined as 2–7 s and 8–13 s after stimulation started, respectively.

For each run, the PSD integral in the desired frequency bands (LFP and multi-unit activity−MUA: 300–3000 Hz) was divided by each band width and *z*-scored before averaging all runs from different animals (Figs. 3E and S10). The "steady-state" power was considered to be the average of power during stimulation after the first second until the end of the stimulation period. The mean "steady-state" power for MUAs and LFPs across animals, Figs. 3F and S8, respectively, were correlated with the mean fMRI "steady-state" signals across animals to further investigate the relationship between these two data modalities. The circles shown in these correlation plots represent the average of each run across animals and the error bars represent the standard deviation of these values. The lowest tested frequency (0.25 Hz) was excluded in this correlation since at such low stimulation frequency, the four individual flashes that occur during the stimulation period are distant enough to be perceived as four distinct stimuli and signals never reach a "steady-state". For the 1 and 2 Hz conditions, the individual flash-induced MUA and LFP power increases are still clearly separated but we assumed the mean power increase to be the "steady-state" power.

### Reporting summary

Further information on research design is available in the Nature Portfolio Reporting Summary linked to this article.

## Data availability

The data that supports the findings described in this paper are available as a figshare repository (https://doi.org/10.6084/m9.figshare.24804948). Source data are provided with this paper.

## Code availability

All the code used to generate the results in this study is available as a figshare repository (https://doi.org/10.6084/m9.figshare.24804948).

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

## Acknowledgements

This study was funded in part by the European Research Council (ERC) (agreement No. 679058), as well as by Fundação para a Ciência e Tecnologia (Portugal), project 275-FCT-PTDC/BBB-IMG/5132/2014. The authors acknowledge the vivarium of the Champalimaud Centre for the Unknown, a facility of CONGENTO which is a research infrastructure co-financed by the Lisboa Regional Operational Programme (Lisboa 2020), under the PORTUGAL 2020 Partnership Agreement through the European Regional Development Fund (ERDF) and Fundação para a Ciência e Tecnologia (Portugal), project LISBOA-01-0145-FEDER-022170. MV thanks Fundação para a Ciência e Tecnologia for a PhD fellowship PD/BD/141560/2018. R.G. thanks Fundação para a Ciência e Tecnologia for a Ph.D. fellowship PD/BD/128297/2017. All authors would like to thank Dr. Alfonso Renart for suggestions and critical reading of the manuscript. The authors would also like to thank Dr. Cristina Chavarrías for the implementation of fMRI acquisition sequences, Ms. Francisca F. Fernandes for customized fMRI analysis MATLAB codes, Dr. Bruno Cruz, Dr. Tiago Monteiro, and Mr. Filipe Rodrigues for input regarding the behaviour experimental design, and Mr. Juan Castiñeiras and Mr. Tiago Costa for advice on data analysis. Finally, the authors would like to thank Ms. Teresa Serradas Duarte for insightful discussions.

## Author contributions

N.S., R.G. and M.V. designed the project and the necessary experiments, and N.S. oversaw the implementation of the project. M.V. acquired and M.V. and R.G. analysed the behavioural data. R.G. acquired and analysed the functional MRI data. R.G. and M.V. acquired and R.G. analysed the electrophysiological data. NS helped in all data analysis. N.S., R.G. and M.V. wrote the paper. All authors discussed the results and implications and commented on the manuscript at all stages.

## Competing interests

The authors declare no competing interests.
