## [Peer Review File · Nature Communications]

Rat superior colliculus encodes the transition between static and dynamic vision modesReviewer #1 (Remarks to the Author):

In this study, authors used a multi-modal approach including behavioural measurement, whole brain fMRI, and local electrophysiology for studying the mechanisms underlying the shift from static to dynamic vision modes in the rat. They found that the fMRI signal in the superior colliculus (SC) transits from positive to negative at the behaviourally measured Flicker Fusion Frequency (FFF) threshold surrogates (18 ± 2 Hz), with a strong linear correlation between fMRI signal and behaviour, while thalamic and cortical visual areas displayed a significantly poorer correlation with the behaviour. This study suggests a critical role for SC in encoding temporal frequency discriminations. This is a very successful study that leverages the advantages of multi-modal approaches to solve an important neuroscience issue. I have a few minor comments that might help further improve the paper.

- 1) Although it is pretty convincing that the SC might be involved in the encoding of temporal frequency discriminations, as it exhibits the largest dynamic range of BOLD changes compared to the LGN and VC, there is a remote chance that the temporal frequency discrimination may occur outside of the visual system. Authors may consider discussing this possibility and examine BOLD signals outside of visual areas.
- 2) Results in Fig. 3 seem well agree with a recent publication (PMID: 36681134).
- 3) The effects of anesthesia need to be discussed in more details. After all, behavioral measurement was conducted in the awake state, and the phenomenon studied is visual perception which should more or less rely on consciousness. Anesthesia may have limited impact on the results of the present study perhaps because the temporal frequency discrimination occurs at low tier visual area (i.e. SC).
- 4) "However, to our knowledge, this is the first clear demonstration of offset peaks at high frequency regimes." Offset response was reported in the paper PMID: 35179203.

Reviewer #2 (Remarks to the Author):

While this study appears interesting and important to understanding the mechanisms of rat visual functions at different temporal frequencies, the authors are advised to address the below issues before the paper can be considered for publication:

- (1) Please provide information about the age and sex of the adult Long Evans rats in each experimental group. Did these demographics differ between behavioral, fMRI, and electrophysiology experiments? Were there any differences in frequency responses over age or between sexes?
- (2) Were there any differences in luminance applied to the rat eyes in the closed settings in fMRI and electrophysiology vs open settings as shown in video? How does luminance/contrast affect frequency responses and FFF thresholds in the pigmented rats?
- (3) How will different oxygenation levels affect behavioral responses, knowing fMRI used higher oxygen levels for maximizing the brain signals?
- (4) Was there any dark or light adaptation in each of the experimental groups?
- (5) Did the VC lesion affect the fMRI image quality at the levels of the SC and LGN? If yes, how could the image quality change affect the fMRI quantitation?
- (6) Will lesioning the VC affect behavioral frequency responses of the awake adult Long Evans rats similar to fMRI under the same behavioral experimental settings?
- (7) How is the visual cortex anatomically connected to the superior colliculus in adult Long Evans rats? Does it connect to the same or different locations in superficial layers in the SC as the retinocollicular projections? If different, would this anatomical difference affect the current fMRI

quantitation?

(8) How does the current experimental setting of light flashing differ from other studies on apparent moving stimuli in evaluating physiological responses in rats (e.g. PMID: 21741483)? Please discuss.

(9) How long did the training last for in the behavioral experiments? Were the rats undergoing fMRI and electrophysiology experiments trained in the same way similar to the separate rats undergoing behavioral experiments for fair comparisons/associations? Could there be any brain response differences in the regions of interests before and after training?

(10) In terms of the discussion of the potential role of SC in novelty detection, could there be any potential contributions/interactions with other non-visual brain regions such as the substantia nigra during awake state and under anaesthesia as in the current behavioral and imaging/electrophysiological studies that may explain the current observations of the onset and offset peaks? (e.g. PMID: 12925855)

(11) It was reported in the supplementary information that 'observed behavioural percentage of reports to the flicker port for the continuous stimulus conditions reaches 10.7% instead of the expected values closer to 0%.'

How did the brain possibly respond during such false positive behavior?

(12) In supplementary information, VC lesioning appeared to induce a much large fluctuation/error bars at 1Hz frequency stimulation than others. Could authors explain why?

(13) line 232: 'show' should be 'shown'

Reviewer #3 (Remarks to the Author):

In this study, the authors studied flicker fusion in the visual cortex, LGN, and SC of rats. They first performed psychophysics and demonstrated that the rats can report flicker at low temporal frequencies. They then performed fMRI on other animals and found that the dependence of imaging responses on temporal flicker was variable across the cortex, LGN, and SC. There was better correlation with behavior for the SC. Then, they performed electrophysiology in the SC and showed flicker responses in the LFP and MUA signals. Finally, they lesioned V1 and imaged the SC. There was a modulation of SC imaging responses at the highest temporal frequency.

The characterisation of flicker fusion in the SC is a valuable contribution, and the paper is generally very clear and understandable. The results are generally convincing. For me, the final experiment with the V1 lesion is still inconclusive. That is, it could be that the SC no longer shows any tracking of the flicker after the V1 lesions, but that there is still some sort of neural response (e.g. a constant steady-state response) that still gives rise to the imaging results. I would remove this final experiment (or move it to the discussion section only), unless the authors can do electrophysiology in the SC during the V1 lesions and still demonstrate the oscillatory responses.

Another general comment is that the writing could be slightly improved, especially in the section of the lesion experiment. In that section, in particular, it felt that the text was switching between past and present tense too frequently.

Below, I provide some more detailed comments that will hopefully be helpful to the authors:

- line 35: I think that a more concrete definition of FFF is required here. Just saying that it defines the transition from static to dynamic modes is not enough. People need to understand the measure as early as possible, and since you define the acronym here, then also give a clearer definition of it at the same time. This is very important in my opinion.

- line 99: the acronym VC was not defined prior to this usage. I presume it means visual cortex,

but this needs to be clarified

- the figures are very hard to see for me. Maybe the PDF conversion reduced their resolution. However, independent of such a resolution issue, I think that the figures could benefit from better clarity. e.g. use longer tick marks, use bigger symbols in the scatter plots, etc.

Also, the titles of all of the figures (above each panel) are inconsistent in their capitalisation. Some titles use title case (e.g. Behavioral Task), but other titles use a weird mixture of capitalisation (e.g. Behavioural reports to Flicker light port). You should be consistent throughout all panels and figures.

- Fig. 2: why were the animals used for the behavior (Fig. 1) not used for the imaging? Right now, panels D, E correlate imaging in one set of animals to behavior in another set of animals. Maybe comment on why the imaging was not done on the same animals.

- Also Fig. 2: I'm confused by the error bars. Panel C says that they represent SEM. However, when I look at the x-axis of panel D, I see very similarly sized error bars, and they are now called SD. What are they? Are they SEM or SD? And if they are SEM, why are they so large?

- lines 145-151: how were the ROI's defined? And were there any other ROI's that were modulated by the stimulation paradigm? i.e. why restrict only to LGN, VC, and SC when the whole point of the imaging was to get brain-wide measures? Maybe there are other regions that were as strongly modulated by the flicker as the SC, but that were not analyzed?

- figure 3B could benefit from an inset with a zoomed in representation of a length of about 1 second, to show the flicker responses at the higher frequencies. I know there might be another figure in supp. Info with some zooming, but I'd like to see a 1-second cycle of the raw signal here and clearly see that the responses are tracking the temporal frequency for the frequencies higher than 1 Hz.

- Figure 3E: why is continuous light suppressing activity in the SC? Isn't the SC a visually-responsive structure that also shows sustained responses to light presence? Also, why not perform single-unit analyses?

- Fig. 4: It seems to me that electrophysiology is absolutely critical in this lesion experiment. That is, it could be that the SC is no longer responding to the flicker with the V1 lesion, but that there is some other kind of neural response (e.g. a steady-state response) that still gives rise to the BOLD signal in fMRI.

- General comment for Introduction and Discussion: please note that flicker fusion in monkey SC neurons was studied in this paper that is not mentioned in the current manuscript:
<https://www.frontiersin.org/articles/10.3389/fncir.2018.00058/full>

Response to Reviewers

We'd like to thank all three Reviewers for their insightful comments, which we were pleased to address in full, and which indeed solidified our previous conclusions and improved its interpretation. Below, we address each comment separately, reproducing the **Reviewer's comment in boldface** followed by our reply underneath. Please note that in some cases, comments may have been broken down and renumbered (labeled **Rx.y** for Reviewer #x comment #y) so that we could ensure that we address each aspect of the comment in full. Text added or modified to the manuscript is shown in blue between quotation marks.

Reviewer #1 (Remarks to the Author):

In this study, authors used a multi-modal approach including behavioural measurement, whole brain fMRI, and local electrophysiology for studying the mechanisms underlying the shift from static to dynamic vision modes in the rat. They found that the fMRI signal in the superior colliculus (SC) transits from positive to negative at the behaviourally measured Flicker Fusion Frequency (FFF) threshold surrogates (18 ± 2 Hz), with a strong linear correlation between fMRI signal and behaviour, while thalamic and cortical visual areas displayed a significantly poorer correlation with the behaviour. This study suggests a critical role for SC in encoding temporal frequency discriminations. This is a very successful study that leverages the advantages of multi-modal approaches to solve an important neuroscience issue. I have a few minor comments that might help further improve the paper.

We'd like to thank the Reviewer for her/his insightful comments.

R1.1 Although it is pretty convincing that the SC might be involved in the encoding of temporal frequency discriminations, as it exhibits the largest dynamic range of BOLD changes compared to the LGN and VC, there is a remote chance that the temporal frequency discrimination may occur outside of the visual system. Authors may consider discussing this possibility and examine BOLD signals outside of visual areas.

We agree. We now added analyses of new regions of interest (ROIs) outside of the visual pathway to test for this remote possibility. In particular, ROIs were delineated in caudate putamen, substantia nigra, ventral tegmental area, parabrachial nucleus, amygdala, hippocampus, anterior cingulate area - all areas did not exhibit any remarkable signals. The new data and analyses were added to SI in **Figure S7**:

Figure S7: Temporal signal profiles in regions of interest outside of the visual pathway. The involvement of other structures outside the visual pathway (caudate putamen, substantia nigra, ventral tegmental area, parabigeminal nucleus, amygdala, hippocampus, anterior cingulate area) with temporal frequency discrimination was investigated. From the obtained time profiles, no clear trend was observed with change of stimulation frequency for any of these regions.

R1.2 Results in Fig. 3 seem well agree with a recent publication (PMID: 36681134).

Thank you for highlighting this work, which in fact we had cited in a previous draft and somehow removed in the submitted version! Indeed the beautiful work by Zhang et al regarding MUA correlations to the BOLD response agrees well with our observation in Figure 3.

This work is now duly discussed and cited (reference number 37) in the revised manuscript (line 261):

“The two signals are highly correlated (also agreeing with recent work (Zhang, Qingqing, et al., NeuroImage, 2023)) and reveal how multiphasic BOLD responses appear to be better characterized by higher frequency band signals such as the MUA (...).”

We note that the presence of cortical pre-ON signals in the referred study were not seen in our work, likely due to the sedated state (compared to awake in Zhang et al).

R1.3 The effects of anesthesia need to be discussed in more details. After all, behavioral measurement was conducted in the awake state, and the phenomenon studied is visual perception which should more or less rely on consciousness. Anesthesia may have limited impact on the results of the

present study perhaps because the temporal frequency discrimination occurs at low tier visual area (i.e. SC).

Thank you for this comment, we agree - this is indeed a confounding factor, although we would propose that our results suggest that the confounding effect is not large given the strong similarities in FFF thresholds. Rather, it is likely that medetomidine sedation in our fMRI and electrophysiology could potentially limit a few aspects of (probably predominantly) cortical responses that are likely more involved in the awake state (as for example in your previous comment).

In this context, it's also interesting to note the (qualitative) similarity between our BOLD responses in the rat visual system and reported awake mouse visual responses with similar frequencies in a study from Dinh, Thi Ngoc Anh, et al. (Dinh, Thi Ngoc Anh, et al., *Neuroimage* 2021), **Figure R1**:

Figure R1: Insets show adapted images from Dinh, Thi Ngoc Anh, et al., *Neuroimage* 2021. The red curves in the insets represent the awake time profiles reported for mice in the referred paper while the yellow and pink time profiles are from our work in medetomidine sedated animals.

This is now more extensively discussed in the Discussion section (line 432):

“First, the sedation state of the animals differed between behaviour (awake) and recordings (fMRI and electrophysiology, where animals were lightly sedated with medetomidine, an $\alpha 2$ agonist (You, Taeyi, Geun Ho Im, and Seong-Gi Kim., *Scientific reports* 2021)) during fMRI and electrophysiology data acquisition. This could potentially account for the very slight differences in perceptual behavioural reports and neural/BOLD signal transitions, although these differences were indeed small. The effects of medetomidine sedation were also shown to be small when

comparing the baseline neural activity to the awake state (Airaksinen, Antti M., et al., *Magnetic resonance in medicine*, 2010) and a (qualitative) comparison with mice visual awake responses (Dinh, Thi Ngoc Anh, et al., *Neuroimage* 2021) shows similarity to those observed in the present work. Furthermore, studies performed in rats using the same dosage of medetomidine as in this work have reported functional connectivity that was partially attributed to wakefulness (Zhao, Fuqiang, et al., *Neuroimage*, 2008). In addition to the good agreement in this study between the measurements from different modalities, this suggests that it is likely that the sedated state induced by medetomidine did not play a critical effect in our observations.”

R1.4 “However, to our knowledge, this is the first clear demonstration of offset peaks at high frequency regimes.” Offset response was reported in the paper PMID: 35179203.

Thank you for pointing this out, we apologize for the oversight and are happy to correct it. Zhang, Qingqing, et al., *Cerebral Cortex* 2022 is now discussed and duly cited (reference number 64) with respect to our findings. Importantly, the work is actually complementary: while the offset responses in Zhang et al were observed as changes from baseline, our paradigm enabled us to directly observe the onset, suppression of activity, and offset in the same response. In addition, in Zhang et al, consistent calcium signals during positive and negative trials suggested that in SC OFF negative trials, BOLD signals did not faithfully reflect the underlying neuronal firing, as the measured neuronal spiking activity was positive while the measured BOLD signal was negative (potentially indicating neurovascular decoupling). In our data, on the other hand, negative BOLD in SC at high stimulation frequency regimes reflected a dramatic decrease in both LFP and MUA power during the stimulation, hence suggesting neuronal suppression.

The manuscript now discusses and cites (reference number 64) Zhang et al in the Discussion section (line 415):

“This study not only confirms the ability of BOLD-fMRI to resolve onset/offset signals previously reported in VC and SC at high frequency stimulation regimes (Zhang, Qingqing, et al., *Cereb Cortex*, 2022 and Zhang, Qingqing, et al., *NeuroImage*, 2023) but also sheds light into a possible behavioural relevance of such signals.”

Reviewer #2 (Remarks to the Author):

While this study appears interesting and important to understanding the mechanisms of rat visual functions at different temporal frequencies, the authors are advised to address the below issues before the paper can be considered for publication:

We thank the Reviewer for her/his comments and suggestions which we feel indeed improved our work.

R2.1 Please provide information about the age and sex of the adult Long Evans rats in each experimental group. Did these demographics differ between

behavioral, fMRI, and electrophysiology experiments? Were there any differences in frequency responses over age or between sexes?

Thank you for this comment - we agree that we have not sufficiently detailed this aspect and we appreciate the opportunity to clarify.

The different groups consist of the following:

- The fMRI control group consisted of 11 female and 7 male rats. Animals were 17.1 ± 8.3 weeks old and weighed 351.4 ± 75.6 g on average.
- The fMRI V1 lesioned group consisted of 10 female and 3 male rats. Animals were 16.4 ± 3.7 weeks old and weighed 375.9 ± 75.1 g on average.
- The fMRI fast group consisted of 5 female and 1 male rats. Animals were 21.0 ± 8.0 weeks old and weighed 332.8 ± 53.0 g on average.
- The electrophysiology control group consisted of 12 female and 6 male rats. Animals were 18.4 ± 7.2 weeks old and weighed 406.4 ± 74.2 g on average.
- The electrophysiology V1 lesioned group consisted of 3 female and 3 male rats. Animals were 19.0 ± 4.1 weeks old and weighed 463.0 ± 106.6 g on average.
- The behavioural group consisted of 7 female rats. Animals were on average 13.7 ± 0.9 weeks old at the start of training and weighed on average 311.8 ± 33.0 g before being water deprived. Due to size constraints of the behavioural setups only female animals could be used since they are smaller than males of identical age. During their time under water deprivation it was ensured they kept increasing in weight and never lost more than 10% of initial weight.

In our experience, in such low-level experiments, males and females perform equally well, and our fMRI results (**Figure R2** shown below) also don't indicate any discernible task-dependent differences between groups of males or females. We cannot completely exclude perhaps very subtle effects between sexes, but those would likely be irrelevant to the tasks at hand.

These important details can be found in the Discussion section (line 443) and in the paper in the Methods section (line 618).

Figure R2: Comparison of male and female fMRI signals. One group composed of males (N=3) and a group composed of females (N=3) were compared for several representative stimulation frequencies. The time profiles confirm the similar temporal evolution of signals for both males and females in the VC, SC and LGN. A one-tailed voxelwise t-test was performed, tested for a minimum significance level of 0.001 with a minimum cluster size of 20 voxels and corrected for multiple comparison using a cluster false discovery rate test (FDR).

R2.2 Were there any differences in luminance applied to the rat eyes in the closed settings in fMRI and electrophysiology vs open settings as shown in video? How does luminance/contrast affect frequency responses and FFF thresholds in the pigmented rats?

Thank you for pointing this out. Studies have shown that the FFF can depend on several characteristics of the stimulus (Simonson, Ernst, and Josef Brozek, *Physiological Reviews*, 1952; Landis, Carney; *Physiological Reviews*, 1954; Williams, Ruth Ann, et al., *Physiology & behavior*, 1985). The two parameters that were kept similar between modalities were the intensity (8.1×10^{-1} W/m²) and the

illuminance (6 - 8 lux) of the stimulus (measured using a photodiode sensor and a luxmeter, respectively). These values were measured at the place where the animal's eyes would be in the MRI and electrophysiology settings and in several places of the behavioral box at nose port level.

The outputs of the LED were always placed in order to binocularly stimulate equally the entire field of view of the animal:

- In the MRI and electrophysiology settings a bifurcated optic fiber connected to an LED was placed with each end in front of each eye (spaced up to 1 cm). In the MRI the LED light is reflected inside the bore while in the electrophysiology the rat's head was covered with a reflective material to ensure that the blue light stimulated the entire field of view of the animal.
- In the behavior box there was one blue LED above the animal's head, placed on the ceiling of the behavioural box, illuminating in a homogeneous way the entire behavioural setup.

Since in all settings the blue light is able to stimulate the entire field of view of the rats, we assume that the differences in optic fiber ends/LED positioning would not affect the results.

These details are now clearly explained in the Methods Section (lines 660 and 734 for behavior and fMRI/electrophysiology, respectively).

R2.3 How will different oxygenation levels affect behavioral responses, knowing fMRI used higher oxygen levels for maximizing the brain signals?

Thank you for pointing out potential effects from the hyperoxia used in fMRI. Most studies looking at effect of hyperoxia on behavioural responses often use physically demanding tasks (Sperlich, B., Zinner, C., Hauser, A. *et al.*, *Sports Med*, 2017; Eynan, Mirit, et al., *Aviation, space, and environmental medicine*, 2003). However, the type of behavior used in this study is of a different kind as it is not physically demanding. Given that hyperoxia in itself is not expected to elicit significant neural activity, we do not expect almost any effect on the measured FFF thresholds, as indeed confirmed by the similarity of our behavioral, fMRI and electrophysiology results.

fMRI results from **Figure S3** show that hyperoxia induces mostly an increase in CNR of the response and not a change in the response timings or shape which is also an indicator that activation/suppression balances in SC are not affected by the increased oxygen concentration.

This is now clearly stated in the Discussion section (line 447).

R2.4 Was there any dark or light adaptation in each of the experimental groups?

This is an important point which we are pleased to be able to address.

In Long Evans rats a minimum of 30 minutes is recommended for complete dark adaptation (Behn, Darren, et al., *Documenta ophthalmologica*, 2003). In our fMRI and electrophysiology groups, we allowed for a minimum of 45 minutes of dark adaptation before data was acquired.

In the behavioural task, the animals were inside the box for approximately 15 minutes before the beginning of the trials, which potentially could have a small effect. To test for this effect, we reanalyzed the behavioural data, but now considering only trials that occurred from the point in which 30 minutes have already elapsed inside the dark box. These new results, shown below and added to the manuscript as **Figure S2**, yielded indistinguishable curves when compared to the full dataset (i.e. with only 15 minutes of light adaptation). The similarity between psychometrics could be due to the animals starting to be more engaged in the task (performing more trials) after 30 minutes inside the box or that 15 minutes is already enough for animals to adapt to the darkness. Therefore, we conclude that adaptation differences due to the slightly shorter time in the behavioural experiments could not have caused any significant bias.

Figure S2: Psychometric curves obtained using only trials starting after either 15 (light blue) or 30 (dark blue) minutes of dark adaptation. Similar results reveal that 15 minutes is already sufficient for the visual system of Long Evans rats to adapt to the dark. The circles correspond to the averaged individual performances.

This discussion was now added to the paper in the supplementary information under the Behavior section and to the main Discussion section as well (line 457).

R2.5 Did the VC lesion affect the fMRI image quality at the levels of the SC and LGN? If yes, how could the image quality change affect the fMRI quantitation?

Thank you for raising this important point. To address this, we calculated the SNR of the control and lesioned fMRI groups. For each animal, one functional scan was used

for the SNR estimation in the SC and LGN. Two ROIs were drawn: one in the structure of interest and another in a noise region containing a similar number of pixels. The SNR was then calculated by dividing the mean signal in the SC/LGN by the mean noise signal. The calculated SNR for the control group in the SC and LGN were 13.3 ± 6.1 and in 9.6 ± 2.6 , respectively, while the calculated SNR for the (V1) lesioned group in the SC and LGN were 13.0 ± 3.5 and 10.8 ± 2.9 , respectively - clearly without any statistically significant differences.

Thus, we can conclude that the V1 lesion did not affect image quality at neither the SC nor LGN levels. This can be also observed from the time profiles shown in **Figure 4** and in **Figure S13** where both the control and lesion groups exhibited similar fluctuations of the mean percent signal change and standard error of the mean.

This information was added to the Results (line 285) and to the Methods section under fMRI Analysis: ROI Analysis (line 813).

R2.6 Will lesioning the VC affect behavioral frequency responses of the awake adult Long Evans rats similar to fMRI under the same behavioral experimental settings?

Thank you for the question. First, it would be prudent to mention that the lesion study was performed to investigate the BOLD responses and how they might be modulated from V1, namely through the gain control effect in the SC - and as such we did not focus on how it could affect behavioral aspects (which to some extent has been investigated in the past). Still, we can draw from the literature to hypothesize about what could be expected. For instance, in humans, VC is clearly necessary for flicker discrimination: complete loss of the VC renders responses to be virtually insensitive to any form of light stimulation. By stark contrast, ablation of the VC in albino rats (Schwartz, A. S. & Clark, G., J Comp Physiol Psychol, 1957) does not impede animals from discriminating flicker from steady light. Histological examination of the LGN bodies in the previously mentioned study showed complete neuronal degeneration, suggesting that visual pathways other than the geniculostriate system (involving LGN and V1) can carry the information needed for the flicker discriminations. This is in-line with our conclusion of SC being the most probable relay point for such information flow with reports of response reorganization in this structure following ablation of VC.

Interestingly, given the strong correlations observed between BOLD/Electrophysiology responses and behaviour, we could at least make an educated guess from the data in **Figure 4**, where lesioning V1 appears to only exert a gain effect in the fMRI SC signal modulation with stimulation frequency. There are several BOLD signal characteristics which can be still observed after V1 lesioning namely the PBR/NBR transition with frequency and the presence of onset/offset peaks at high frequency regimes. From the characteristics we can hypothesize that the measured FFF threshold would not be expected to be dramatically affected by the V1 lesions. If anything, assuming that the PBR/NBR transition can be used as a proxy for the FFF threshold surrogate, the measured thresholds could be slightly shifted to higher frequencies due to the upward regulation of negative responses.

Given this aspect goes well beyond the scope of our work, we decided to keep this discussion only in this Response Letter but if the Reviewer deems it critical, we can certainly include the discussion in the paper before publication.

R2.7 How is the visual cortex anatomically connected to the superior colliculus in adult Long Evans rats? Does it connect to the same or different locations in superficial layers in the SC as the retinocollicular projections? If different, would this anatomical difference affect the current fMRI quantitation?

Thank you for providing us with the opportunity to explain this aspect. The superficial layers of SC receive topographic inputs from retinal ganglion cells (RGCs) and corticotectal feedback from the visual cortex. In rats, it has been shown that most cortical inputs target non-GABAergic cells in the superficial SC (Boka et al. 2006).

A study performed in mice has shown that cortical inputs are topographically organized and aligned with retinal inputs (Triplett et al., Cell, 2009). Results from this work suggest an instructive role of RGCs in the mapping and alignment of cortical visual projections to the SC. Spontaneous activity occurring prior to eye opening appears to be key for the alignment of visual maps and proper lamination of corticocollicular projections. Other studies have suggested a similar mechanism in barn owls and ferrets for visual input in auditory mapping in the midbrain (King et al., 1988; Knudsen and Brainard, 1991) therefore hinting at a general model in which convergent inputs to a central structure use coincidental activity to achieve alignment with the primary map. The same alignment between retinal and cortical inputs in the SC has been described in the rat (LeVere, T. E., 1978).

Furthermore, studies have shown superficial SC neurons inherit basic feature selectivity from the retina and modulate their response magnitude depending on cortical input (Liang et al., Neuron, 2015) which is in-line with the map alignment mechanisms suggested above.

We have added a brief discussion on these connections and their agreement with our results in the Discussion section (line 397).

R2.8 How does the current experimental setting of light flashing differ from other studies on apparent moving stimuli in evaluating physiological responses in rats (e.g. PMID: 21741483)? Please discuss.

Thank you for mentioning this interesting study, which we cite (reference 26) in our work too (Lau, Condon, et al., Neuroimage, 2011). In this fMRI study, Lau, Condon, et al. used a visual set-up consisting of four light spots arranged in a linear array. The LEDs would turn ON and OFF sequentially at different rates to create five effective illusory speeds of motion and the stimulus “speed” dependence on visual pathway signals was investigated. The percent BOLD signal increased for the first four speeds while there was a reduction for the highest speed in both SC and LGN.

Indeed, both studies investigated induced visual illusions due to increased frequency of stimulus; however, while ours is solely manipulating the temporal domain, the mentioned study creates an illusory “motion” in the spatial dimension. This is an important difference between the two studies since when variables such as “motion”

and “spatial frequency” are introduced, the behavioral relevance to the stimulus and the way it is processed along the visual pathway might be completely different.

Take, for example, the shape of the obtained fMRI responses. Although a modulation is observed for higher speeds, the shape is very different from the fMRI responses measured in our study. Responses in the mentioned study appear to show a slight increase in percent BOLD change during the stimulation on period while in our case there is an initial signal peak followed by a slight decrease and a steady-state along with onset/offset peaks for high frequency regimes.

Furthermore, the existence of four spatially separated LEDs, 7 cm away from the rat's eyes, can induce several differences such as eye movement to follow the stimuli or even stimulation of different regions of the rat's field of view. On the other hand, in our case the LEDs were placed close to the eye to homogeneously stimulate the entire field of view of the animals.

In the Discussion section of the mentioned work the authors alert for the comparison of this study with stationary flicker studies such as ours: “The rat brain regions activated by stationary flicker and moving stimulation appear to agree, as this study also noted SC, LGN, and visual cortex responses at multiple speeds. However, we caution that stationary flicker and motion comparison studies should be done together under the same experimental conditions. Also, greater differences may be present in cortical areas, which have not been examined with sufficiently high spatial resolution in rat visual fMRI studies.”

In summary, our work isolates the temporal aspects of visual continuity effect, as well as investigates it from behavioural, fMRI and electrophysiological perspectives.

The distinction between studies is now briefly mentioned in line 36.

R2.9 How long did the training last for in the behavioral experiments?

Thank you for allowing us to clarify this.

The training of the animals in the general contingency of the task – where only two frequencies (2Hz and continuous light) were presented - took on average 3.0 ± 1.0 weeks while the assessment of their FFF threshold surrogate– where more frequencies, including probe frequencies, were presented - took place over 3.8 ± 0.4 weeks.

This is now mentioned in the Methods section under the Behavior section (line 683).

R2.10 Were the rats undergoing fMRI and electrophysiology experiments trained in the same way similar to the separate rats undergoing behavioral experiments for fair comparisons/associations?

Thank you for pointing out that this is not sufficiently clear.

In all three modalities, animals were fully naïve to the presented stimuli prior to the procedure. This was done in order to accomplish a fair comparison between groups and because we aimed at probing the natural FFF threshold of the animals. For this reason, and to not influence the measured thresholds, we (1) did not train animals prior to MRI or electrophysiological experiments and (2) trained the animals in the behaviour task for a relatively short period of time, in an attempt to prevent behavioural strategies not directly related to their perception of the light, from skewing the results.

Therefore, the percept originated by the stimuli should be similar in all groups of animals, at least at the level of the feed-forward signal. We cannot completely exclude subtle changes due to feedback signals originating in associative areas that respond to learned processes such as our task, but we would not necessarily expect significant changes at the level of the SC.

This is now mentioned in our manuscript in the Methods section (line 627)

R2.11 Could there be any brain response differences in the regions of interests before and after training?

Thank you for this question - which we are very interested in ourselves very much. Clearly, addressing it would require a whole, full-blown, plasticity-based study of its own. While this learning aspect is completely outside the scope of our work, we can share that we do have some preliminary data that indeed shows some fMRI-based effects due to training (i.e. animals scanned before being trained, and at the end of the training sessions). We will be sure to report these results in due course, once we complete this full-blown study.

R2.12 In terms of the discussion of the potential role of SC in novelty detection, could there be any potential contributions/interactions with other non-visual brain regions such as the substantia nigra during awake state and under anaesthesia as in the current behavioral and imaging/electrophysiological studies that may explain the current observations of the onset and offset peaks? (e.g. PMID: 12925855)

Thank you for pointing out this work. New regions of interest outside of the visual pathway, including substantia nigra and ventral tegmental area, were investigated and represent now **Figure S7**. Although the work of Eliane Comoli et al., reveals a retino-tecto-nigral circuit where SC evoked potentials elicit and modulate nigral evoked potentials, in our delineated ROIs neither clear trends nor onset/offset signals were observed suggesting that these structures are not involved in temporal frequency discrimination. Several aspects could be involved in the lack of responses in these dopaminergic areas, namely the usage of different anesthetics/sedatives (urethane vs medetomidine). This can be further investigated; however, it is out of the scope of this work.

Figure S7: Time profiles for regions of interest outside the visual pathway. The involvement of other structures outside the visual pathway (caudate putamen, substantia nigra, ventral tegmental area, parabigeminal nucleus, amygdala, hippocampus, anterior cingulate area) with temporal frequency discrimination was investigated. From the obtained time profiles, no clear trend was observed with change of stimulation frequency for any of these regions.

R2.13 It was reported in the supplementary information that 'observed behavioural percentage of reports to the flicker port for the continuous stimulus conditions reaches 10.7% instead of the expected values closer to 0%.'

How did the brain possibly respond during such false positive behavior?

Thank you for raising this point. Our hypothesis is that the stimulus representation at the level of the SC is not significantly modulated by experience, or by the animal learning the lateral associations of the task (also mentioned in point number 10). However, this is not necessarily true for other areas of the brain and it is possible that the task elicits cortical plasticity, especially in up-stream associative areas of the brain (Garvert et al., *Neuron*, 2015; Herdener et al., *Journal of Neuroscience*, 2010; Yang and Maunsell, *Journal of Neuroscience*, 2004; Lee et al., *Nature Neuroscience*, 2002). Such areas might adapt to the contingencies of the task and the animals may try to optimize their strategy. In this case, if uncertain of the stimulus, it might be more profitable to respond to the flicker port - the trials are even to both sides in terms of amount and the frequencies close or above the FFF threshold are rare, but there are more flicker frequencies and only one continuous.

This is now clearly mentioned in the supplementary Behavior discussion.

R2.14 In supplementary information, VC lesioning appeared to induce a much large fluctuation/error bars at 1Hz frequency stimulation than others. Could authors explain why?

Thank you for so astutely catching this visual mistake in our figure due to an error in the graphic software layers! This is now corrected. The signal fluctuations are now similar for all the frequencies in both groups of animals. Please see below the corrected **Figure S13**.

Figure S13: Cortical and thalamic fMRI temporal profiles after V1 ibotenic lesion (mean \pm s.e.m. across animals). V1 profiles appear flat as expected from localized ibotenic acid lesions while LGN profiles appear similar to the control conditions with clear modulation as stimulation frequency increases but never reaching negative values.

R2.15 line 232: 'show' should be 'shown'

Corrected, thank you (now line 280).

Reviewer #3 (Remarks to the Author):

In this study, the authors studied flicker fusion in the visual cortex, LGN, and SC of rats. They first performed psychophysics and demonstrated that the rats can report flicker at low temporal frequencies. They then performed fMRI on

other animals and found that the dependence of imaging responses on temporal flicker was variable across the cortex, LGN, and SC. There was better correlation with behavior for the SC. Then, they performed electrophysiology in the SC and showed flicker responses in the LFP and MUA signals. Finally, they lesioned V1 and imaged the SC. There was a modulation of SC imaging responses at the highest temporal frequency.

The characterisation of flicker fusion in the SC is a valuable contribution, and the paper is generally very clear and understandable. The results are generally convincing. For me, the final experiment with the V1 lesion is still inconclusive. That is, it could be that the SC no longer shows any tracking of the flicker after the V1 lesions, but that there is still some sort of neural response (e.g. a constant steady-state response) that still gives rise to the imaging results. I would remove this final experiment (or move it to the discussion section only), unless the authors can do electrophysiology in the SC during the V1 lesions and still demonstrate the oscillatory responses.

Thank you for your insights and the effort in reviewing our work.

R3 General: Another general comment is that the writing could be slightly improved, especially in the section of the lesion experiment. In that section, in particular, it felt that the text was switching between past and present tense too frequently.

Thank you very much, we have improved the language throughout, as suggested.

Below, I provide some more detailed comments that will hopefully be helpful to the authors:

R3.1- line 35: I think that a more concrete definition of FFF is required here. Just saying that it defines the transition from static to dynamic modes is not enough. People need to understand the measure as early as possible, and since you define the acronym here, then also give a clearer definition of it at the same time. This is very important in my opinion.

We agree and thank you for pointing this out. In response, we have extended the definition of the FFF in line 37:

“In the temporal continuity illusion, the Flicker Fusion Frequency (FFF) threshold is typically defined as the frequency at which the transition from static to dynamic vision modes occurs as reported by the specific experimental modality being used, which can lead to some level of ambiguity. In electrophysiology the FFF threshold has been defined as the frequency at which individual flash evoked potentials can no longer be resolved, while in behavioural essays the threshold is defined based on animals’ reports on their visual perception.”

R3.2 line 99: the acronym VC was not defined prior to this usage. I presume it means visual cortex, but this needs to be clarified

Thanks for noticing - we corrected as requested. The definition of VC now appears explicitly in the introduction (line 29):

“By contrast, how visual systems resolve luminance changes over time has yet to be explained on a systems level, with most studies focusing mainly on the retina and/or the visual cortex (VC). “

R3.3- The figures are very hard to see for me. Maybe the PDF conversion reduced their resolution. However, independent of such a resolution issue, I think that the figures could benefit from better clarity. e.g. use longer tick marks, use bigger symbols in the scatter plots, etc.

Also, the titles of all of the figures (above each panel) are inconsistent in their capitalisation. Some titles use title case (e.g. Behavioral Task), but other titles use a weird mixture of capitalisation (e.g. Behavioural reports to Flicker light port). You should be consistent throughout all panels and figures.

Thank you for helping us to make the Figures more readable. The Figures have been edited accordingly and indeed as you suspected, quality had been lost previously when converting to PDF. We will ensure that high resolution tiffs are reproduced during production of the article.

R3.4- Fig. 2: why were the animals used for the behavior (Fig. 1) not used for the imaging? Right now, panels D, E correlate imaging in one set of animals to behavior in another set of animals. Maybe comment on why the imaging was not done on the same animals.

Thank you for allowing us to clarify this. A similar point was raised by Reviewer 2 point 10. In short - we did not want to investigate effects that could arise from learning and/or plasticity. As the animals learn the task contingency an association between the light stimulus, a lateralized port and a reward is established. This could lead to changes in the brain areas involved in the response to the flashing stimulus. For example, higher-order associative cortical areas and dopaminergic areas related to the expected reward could start to be involved once the stimulus is presented after the animals performed the task. In this work we focused on the naïve brain, and therefore decided to use different animals in the different modalities. For each experiment the animals were presented with the light stimulus for the first time.

This is now explicitly clarified in the Methods section (line 627).

R3.5- Also Fig. 2: I'm confused by the error bars. Panel C says that they represent SEM. However, when I look at the x-axis of panel D, I see very similarly sized error bars, and they are now called SD. What are they? Are they SEM or SD? And if they are SEM, why are they so large?

Thank you for pointing out this confusion. In fact, the report is correct: In **Figure 2C** the error bars represent the standard error of the mean (as appropriate for this cohort) while in **Figure 2D** the correlations are computed using the mean values of steady-state in each run across animals for each condition (i.e. stimulation frequency) and the error bars represent the standard deviation of these values (as appropriate again).

This is now explicitly explained in the Methods section under fMRI Analysis: ROI Analysis (line 800).

R3.6- lines 145-151: how were the ROI's defined?

We used anatomical areas defined a-priori (i.e. not through t-maps) according to the 6th Edition of Paxinos & Franklin's rat brain atlas for manual ROI delineation.

This is now explicitly stated in the Methods section under fMRI Analysis: ROI Analysis (line 786).

R3.7- And were there any other ROI's that were modulated by the stimulation paradigm? i.e. why restrict only to LGN, VC, and SC when the whole point of the imaging was to get brain-wide measures? Maybe there are other regions that were as strongly modulated by the flicker as the SC, but that were not analyzed?

Thank you for pointing this out, as did the other Reviewers. In response, we now drew new ROIs outside of the visual pathway and added the results to the analysis. The results can be found in our new **Figure S7**, where the time profiles clearly show no interesting trends that suggest involvement of any of these structures with temporal frequency discrimination.

Figure S7: Time profiles for regions of interest outside the visual pathway. The involvement of other structures outside the visual pathway (caudate putamen, substantia nigra, ventral tegmental area, parabigeminal nucleus, amygdala, hippocampus, anterior cingulate area) with temporal frequency discrimination was investigated. From the obtained time profiles, no clear trend was observed with change of stimulation frequency for any of these regions.

R3.8- Figure 3B could benefit from an inset with a zoomed in representation of a length of about 1 second, to show the flicker responses at the higher frequencies. I know there might be another figure in supp. Info with some zooming, but I'd like to see a 1-second cycle of the raw signal here and clearly see that the responses are tracking the temporal frequency for the frequencies higher than 1 Hz.

We agree and have thus added the requested inset to **Figure 3B**. The zoomed plots show how for higher frequencies, even 25 Hz, neural activity can still follow the individual light flashes although with dramatic amplitude decreases. The added caption for this inset is highlighted in yellow, and we thank the Reviewer again for making our Figure much better with this excellent suggestion!

Figure 3B: Median LFP traces. LFP traces show individual flash-induced LFP oscillations for the 1 Hz condition and onset and offset peaks for the higher frequencies. Zoomed LFP plots show the first second of responses where flash induced LFP (with increasingly reduced amplitude) can be observed until the 25 Hz stimulation regime.

R3.9- Figure 3E: why is continuous light suppressing activity in the SC? Isn't the SC a visually-responsive structure that also shows sustained responses to light presence?

Thank you for the comment. One interpretation could be that when individual flashes are no longer perceptible, the entire stimulation period is “fused” and integrated as “one long flash” and the SC onset/offset peaks reflect brightness changes or novelty detections at the edges of stimulation – from dark to bright (onset peak) and from bright to dark (offset peak).

In **Figure 4**, we propose the different contributions to the SC signals occurring in the visual pathway. Within SC itself we consider there are two effects overlapping: (i) a “novelty detection” effect (onset/offset signals) and (ii) “frequency discrimination” perception – a constant effect of activation/suppression of SC activity modulated by stimulation frequency. When SC can still follow individual flashes (at low frequencies), each flash represents a novelty event and therefore single flash induced activations are observed. On the other hand, when SC neurons are no longer capable of regaining their full excitability in-between light flashes (for high frequencies when the inter-stimulus interval is reduced), the stimulation period becomes an illusory continuous light with no novel events occurring while the light is “on”. In the latter scenario the novelty events only occur when the stimulus starts and once again when the stimulus finishes, corresponding to the two peaked signals measured for higher frequencies. In addition a cortical gain control also takes place and modulates the SC responses – when frequencies are low it potentiates the evoked responses, and when frequencies are high, it increases the suppression in SC to avoid instigation of the novelty detection.

These aspects are clarified in the Results section (line 296) and in the Discussion section (line 422).

R3.10 Also, why not perform single-unit analyses?

Thanks for the comment. Our electrophysiological experiment was designed to support the fMRI data, to ensure that the modulations observed are of neural origin (a common criticism of BOLD responses is that effects may be disconnected from the neural origins via the vascular “filter”). As such, we were mainly interested in population level activity, clearly reflected in the MUA and LFPs - which are almost always what accompanies fMRI work with electrophysiology. We defer single unit analyses to future research that is more interested in specific mechanisms relating to excitation and inhibition and the data will be made freely available in an online repository for researchers interested in these aspects.

R3.11- Fig. 4: It seems to me that electrophysiology is absolutely critical in this lesion experiment. That is, it could be that the SC is no longer responding to the flicker with the V1 lesion, but that there is some other kind of neural response (e.g. a steady-state response) that still gives rise to the BOLD signal in fMRI.

We completely agree and thank the Reviewer for pushing us to perform these extra tricky experiments. In response, we added an extra group of 6 animals (3 males and

3 females) to our study with V1 lesions and recorded electrophysiological signals in the SC. Importantly, the results from this lesioned group (**Figure 4D**) were very similar to those obtained from the control group (**Figure 3**) indicating that the lesion performed in V1 did not affect the capability of SC to follow the stimulus nor hindered the observed modulation of SC activity with different stimulation frequencies. Please see the new **Figure 4D** below. This reinforces our conclusions and solidifies them.

Figure 4D: Electrophysiology results. Electrophysiological signals were recorded in lesioned animals to confirm the presence of the SC oscillatory responses. From the left and middle plots, single flash evoked responses are observed until 25 Hz. Onset and offset signals are still observable with the latter only present for the higher stimulation conditions. MUA power plots show a power reduction with stimulation frequency.

We further added the following to the paper in the Results section (line 289):

“To confirm that SC could still follow individual flashes in the V1 lesioned-animals as it would in the control conditions, electrophysiological responses were recorded in 6 animals with lesions in V1 (**Figure 4D**). The trends strongly resemble those of the control group, with individual-flash evoked responses still visible at 25 Hz. MUA power plots (Figure 4D rightmost panel), also exhibited reductions in power during the stimulation period for higher stimulation frequencies as in control conditions (**Figure 3E**). In addition, a MUA power reduction below baseline can be observed already at 25 Hz stimulation, as was observed for controls.”

R3.12 General comment for Introduction and Discussion: please note that flicker fusion in monkey SC neurons was studied in this paper that is not mentioned in the current manuscript:

<https://www.frontiersin.org/articles/10.3389/fncir.2018.00058/full>

Thank you for pointing out this important reference which we neglected to cite in the first round- This paper is now referenced as requested - reference number 43.

Reviewer #1 (Remarks to the Author):

Authors addressed all my comments. I have no additional concern and recommend it for publication in Nature Communications.

Reviewer #2 (Remarks to the Author):

The authors have adequately addressed this reviewer's comments.

Reviewer #3 (Remarks to the Author):

My comments were addressed.

Figures are still not perfectly good in my opinion. tick marks on x-axes are tiny and lines are sometimes too thin, and labels too small in some places.

We would like to thank all three Reviewers for recommending publication of our work in Nature Communications.

As to the last remaining comment, from Reviewer #3, regarding the tick marks and labels on the figures – this has now been addressed as well. Tick marks were made larger, label font was increased. We have also made efforts for the overall quality of the images to be the highest possible and will ensure in the proof stage that all images are with the highest standard of quality.